# Future Link Prediction
# Without Memory or Aggregation

**Lu Yi[1], Runlin Lei[1], Fengran Mo[2], Yanping Zheng[1],* Zhewei Wei[1]***
[1]Renmin University of China, [2]Université de Montréal
{yilu, runlin_lei, zhengyanping, zhewei}@ruc.edu.cn
fengran.mo@umontreal.ca

**Yuhang Ye[3]**
[3]Huawei Poisson Lab, Huawei Technology Ltd.
yeyuhang@huawei.com

## Abstract

Future link prediction on temporal graphs is a fundamental task with wide applicability in real-world dynamic systems. These scenarios often involve both recurring (seen) and novel (unseen) interactions, requiring models to generalize effectively across both types of edges. However, existing methods typically rely on complex memory and aggregation modules, yet struggle to handle unseen edges. In this paper, we revisit the architecture of existing temporal graph models and identify two essential but overlooked modeling requirements for future link prediction: representing nodes with unique identifiers and performing target-aware matching between source and destination nodes. To this end, we propose Cross-Attention based Future Link Predictor on Temporal Graphs (CRAFT), a simple yet effective architecture that discards memory and aggregation modules and instead builds on two components: learnable node embeddings and cross-attention between the destination and the source's recent interactions. This design provides strong expressive power and enables target-aware modeling of the compatibility between candidate destinations and the source's interaction patterns. Extensive experiments on diverse datasets demonstrate that CRAFT consistently achieves superior performance with high efficiency, making it well-suited for large-scale real-world applications.

## 1   Introduction

Dynamic systems are typically formulated as temporal graphs, where nodes and edges represent entities and their interactions, and each edge is annotated with a timestamp [10, 15, 47]. The prediction of future interactions between entities is defined as a future link prediction task in existing studies [9], which is crucial and widely applied in real-world systems [3, 28], such as social networks [8, 11] and collaboration platforms [33]. Future interactions can be categorized into repeated interactions with historical neighbors and new interactions with previously unconnected nodes [44]. For instance, in social networks, individuals often maintain regular contact with close friends while occasionally reaching out to other acquaintances. Similarly, in academic collaboration networks, researchers may frequently publish with long-term collaborators while also initiating projects with new coauthors. Thus, to precisely predict both seen and unseen interactions is important to enhance user experience in different dynamic systems and improve their utility. This dual nature poses a nontrivial challenge for future link prediction, demanding models that generalize across both seen and unseen edges.

---

*Yanping Zheng and Zhewei Wei are the corresponding authors.

39th Conference on Neural Information Processing Systems (NeurIPS 2025).

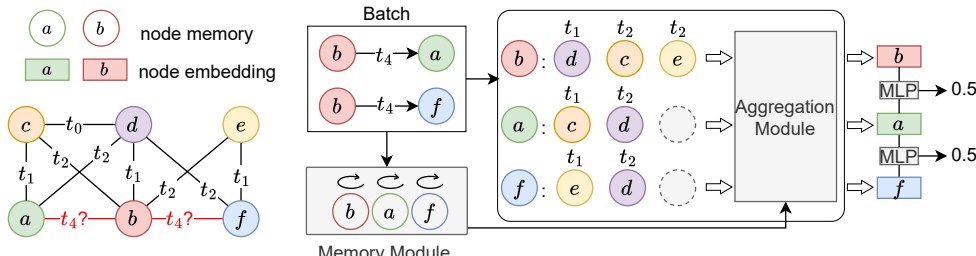

Figure 1: The general model architecture of existing temporal graph learning methods.

Existing temporal graph learning methods perform future link predictions by adopting one or both of the memory and aggregation modules [30, 43], as illustrated in Figure 1. The memory module represents node interaction history in a compressed memory state, while the aggregation module generates node embeddings by aggregating historical neighbors. For example, TGN [30] uses a GRU [6] to update node memory and a graph attention layer for aggregation, whereas DyGFormer [46] omits the memory module and directly applies self-attention over historical neighbors. Although equipped with a sophisticated architecture, these methods suffer substantial performance degradation on recent benchmarks like TGB-Seq [44], which emphasize unseen edge prediction in real-world scenarios, thus limiting their practical utility. This consistent underperformance may stem from a deeper architectural misalignment with the core demands of future link prediction, prompting a critical question: *Are memory and aggregation modules truly effective for future link prediction, especially given the need to handle both seen and unseen edge scenarios?*

To investigate this question, we identify two essential model capabilities that are missing from the existing architecture but are key to addressing the challenges of future link prediction. **First**, the memory and aggregation modules have limited expressive power due to the absence of *unique node identifiers*. They typically operate over historical interaction data, which often lack distinctive node or edge features [13, 17, 27, 44]. Therefore, models often struggle to distinguish between nodes using only their edge timestamps [44]. This limitation becomes particularly severe in datasets with a low proportion of seen edges, where the model must rely on a more expressive representation of historical interactions to generalize to unseen edges. **Second**, both modules are generally applied to individual nodes in isolation, without modeling how well the candidate destination (i.e., the *target*) aligns with the source's recent interaction patterns, a capability we refer to as *target-aware matching*. Although some recent methods [46, 38, 21, 5] leverage correlation encodings to capture the relation between the source and destination, they primarily focus on the connections between their neighborhoods, rather than directly assessing semantic compatibility between the destination and the source's behavioral history. Therefore, the framework designs of these methods are not well aligned with the core demands of future link prediction – to evaluate how well a destination fits the source context.

In this paper, we introduce a minimalistic architecture that centers on these two essential modeling requirements: representing nodes with unique node identifiers and performing target-aware matching, while discarding the commonly used memory and aggregation modules. Specifically, we propose **CRoss-Attention based Future Link Predictor on Temporal Graphs (CRAFT)**, which employs learnable embeddings to serve as node identifiers and a cross-attention mechanism between each candidate destination and the source's recent neighbors to realize target-aware matching. By integrating these two components, our CRAFT provides strong expressive power and enables direct evaluation of the destination's compatibility with the source's interaction patterns, making it effective for both seen and unseen edge prediction. Overall, our main contributions are as follows:

- We propose CRAFT, a simple yet effective architecture for temporal graph learning. Unlike existing models that rely on memory and aggregation modules, CRAFT is built on two essential components tailored for future link prediction: unique node identifiers and target-aware matching.

- With strong expressiveness and destination-conditioned context modeling, CRAFT effectively addresses both seen and unseen edge prediction, addressing the limitations observed in prior work.

- Extensive experiments on 17 datasets, including the TGB and TGB-Seq benchmarks, demonstrate that CRAFT consistently delivers superior performance across both types of prediction tasks while maintaining high efficiency, making it well-suited for large-scale real-world applications.

## 2  Related Work

**Temporal graph learning methods for future link prediction.** Temporal graph learning focuses on modeling the evolving patterns of node interactions over time, with future link prediction serving as a core task [32]. TGN [30] initially propose the memory-aggregation architecture for temporal graph learning and categorize earlier methods within this architecture, including JODIE [17], DyRep [35], and TGAT [42]. Among them, JODIE, DyRep, and TGN adopt RNNs [40] in the memory module to update node memory when new edges arrive, and utilize time projection, identity function, and temporal graph attention to aggregate historical information, respectively. TGAT and many subseqent methods generate time-dependent node representations directly by aggregating historical neighbors without the memory module. These methods leverage various techniques to model dynamic interactions, including the MLP-Mixer [34] in GraphMixer [7], attention mechanisms in TGAT [42], DyGFormer [46], TCL [37], and SimpleDyG [41], as well as temporal point processes (e.g., Hawkes processes [12, 24]) in TREND [39] and LDG [16], etc.

**Limitations of existing methods on predicting unseen edges.** Poursafaei et al. [27] are the first to observe the imbalance between seen and unseen edges in existing datasets. They propose a simple heuristic method, EdgeBank, which memorizes historical edges without any learnable components, yet achieves surprisingly strong performance on many benchmarks due to their high ratio of seen edges. To make evaluation more challenging, Huang et al. [13] introduce the TGB benchmark, featuring large-scale datasets and rigorous multiple negative sampling strategy, while some datasets still exhibit excessive seen edges. To fill this gap, Yi et al. [44] propose the TGB-Seq benchmark, designed to evaluate performance on datasets with minimal seen edges. Their findings further revealed that most existing methods struggle to generalize to unseen edge. This persistent performance gap between seen and unseen edges motivates us to rethink the architecture of temporal graph learning.

## 3  Problem Formulation

**Temporal Graphs.** We define a temporal graph as $G = (V, E)$, where $V$ is the set of nodes, and $E$ is a sequence of edges ordered by non-decreasing timestamps, i.e., $E = \{e_1, e_2, \cdots, e_m\}$, with each edge $e_i = (u_i, v_i, t_i)$ denoting an interaction from the source node $u_i$ to the destination node $v_i$ at time $t_i$. This graph model is also known as a "link stream" or "streaming graph" [2, 18, 22]. Throughout this paper, we use the terms *target* and *destination* interchangeably.

**Future Link Prediction.** Future link prediction aims to estimate the likelihood of an interaction between a source node $u$ and a destination node $v$ at a future time $t$, given all historical edges before $t$ in the graph $G$. We follow prior works [13, 44] to frame this task as a ranking problem, as real-world applications often require identifying the most likely destination from a large pool of candidates. Accordingly, the objective is to rank the positive destination node $v$ higher than sampled negative candidates, conditioned on the source node $u$ and the prediction time $t$.

**Seen and Unseen Edge Prediction.** In this paper, we define *seen edges* as edges that have appeared before the prediction time, and *unseen edges* as edges that have not appeared before the prediction time. Both seen and unseen edge predictions are common in real-world applications. For instance, users on an e-commerce platform may regularly purchase routine products (seen edges) while occasionally buying new items such as clothing (unseen edges). As noted in prior work [44], existing methods perform well on seen edge prediction but struggle with unseen edge prediction. Our work addresses both settings, bridging an important gap in the literature, which has predominantly focused on seen edges.

## 4  Proposed Methods

In this section, we first introduce how our CRAFT's architecture achieves two essential modeling requirements for future link prediction (Section 4.1): 1) representing nodes with unique identifiers to enhance the model expressive power and 2) performing target-aware matching to enable direct assessment of the temporal compatibility between the source and destination. Then, we details on the necessity of these two components in Section 4.2. In Section 4.3, we analyze the time complexity of CRAFT to demonstrate its superior efficiency and compare it with existing methods.

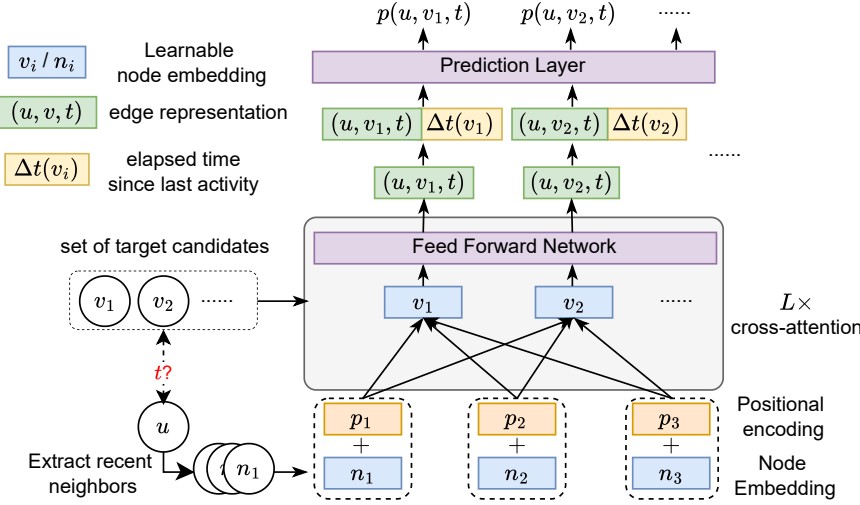

Figure 2: Model architecture of our proposed method, CRAFT: representing nodes with learnable embeddings and performing target-aware matching using cross-attention.

## 4.1 Model Architecture

The architecture of CRAFT is shown in Figure 2. Given the source $u$ and a set of candidate destinations $D = \{v_1, v_2, \cdots, v_q\}$ at time $t$, we first extract the $k$ recent neighbors of $u$ before $t$, and look up node embeddings for these neighbors and the destinations. For the $i$-th neighbor $n_i$ of $u$, we add positional encoding to the node embedding to capture the temporal order of these neighbors. Then we adopt a cross-attention module between the candidate destinations and the source's neighbors to compute the edge representations. To capture the temporal status of the destinations, we encode the elapsed time since last activity of these destinations and concatenate it with the edge representations. Finally, we use an MLP to predict the edge likelihoods. We detail the process as follows.

**Learnable node embedding as unique identifiers.** To represent nodes with unique identifiers, we introduce learnable node embeddings that are trained jointly with the model. Such trainable embeddings can encode the contextual information from interaction history of nodes, thereby serving as the unique latent identifiers. Specifically, we denote the embedding of node $n$ as $\mathbf{e}(n) \in \mathbb{R}^d$. For the $i$-th neighbor $n_i$ of the source $u$, we add positional encoding $\mathbf{p}(i)$ to the node embedding $\mathbf{e}(n_i)$ to represent the temporal order of these neighbors. The updated embedding is given by:

$$\bar{\mathbf{e}}(n_i) = \mathbf{e}(n_i) + \mathbf{p}(i), \tag{1}$$

where $\mathbf{p}(i)$ is also trainable. The positional encoding enable the model to capture the short-term and long-term interests of the source, which affect the future interactions crucially. Then, we obtain the embedding sequence of the source's neighbors: $\mathbf{S} = [\bar{\mathbf{e}}(n_1), \bar{\mathbf{e}}(n_2), \cdots, \bar{\mathbf{e}}(n_k)]$ and the embedding sequence of the candidate destinations: $\mathbf{D} = [\mathbf{e}(v_1), \mathbf{e}(v_2), \cdots, \mathbf{e}(v_q)]$. Note that these destination nodes are independent of each other, and we concatenate them only for parallel computation.

**Target-aware matching via cross-attention.** Considering the inherent requirement of future link prediction to assess the destination's compatibility with the source's historical interaction patterns, we implement target-aware matching through cross-attention between the candidate destinations and the source's neighbors. This mechanism allows each candidate destination to directly attend to the source's historical interactions, thus better evaluating the alignment between the destination and the source's temporal preferences. We define the Attention and FFN function [36] as follows:

$$\text{Attention}(\mathbf{Q}, \mathbf{K}, \mathbf{V}) = \text{softmax}(\frac{\mathbf{Q}\mathbf{K}^\top}{\sqrt{d}})\mathbf{V}, \tag{2}$$

$$\text{FFN}(\mathbf{H}) = \text{GELU}(\mathbf{H}\mathbf{W}_1 + \mathbf{b}_1)\mathbf{W}_2 + \mathbf{b}_2. \tag{3}$$

Our cross-attention module is stacked with multiple layers, and each layer includes a multi-head attention network (MHA) and a feed-forward network (FFN). The $\ell$-th layer calculation is given by:

$$\mathbf{Z}_i^{(\ell)} = \text{Attention}(\mathbf{H}^{(\ell-1)}\mathbf{W}_{\ell,i}^Q, \mathbf{SW}_{\ell,i}^K, \mathbf{SW}_{\ell,i}^V), \tag{4}$$

$$\mathbf{Z}^{(\ell)} = \text{MHA}(\mathbf{H}^{(\ell-1)}, \mathbf{S}) + \mathbf{H}^{(\ell-1)} = \text{concat}\left(\mathbf{Z}_0^{(\ell-1)}, \mathbf{Z}_1^{(\ell-1)}, \cdots, \mathbf{Z}_{h-1}^{(\ell-1)}\right)\mathbf{W}_o + \mathbf{H}^{(\ell-1)}, \tag{5}$$

$$\mathbf{H}^{(\ell)} = \text{FFN}(\mathbf{Z}^{(\ell)}) + \mathbf{Z}^{(\ell)}. \tag{6}$$

where $\mathbf{W}_{\ell,i}^Q \in \mathbb{R}^{d \times d_h}$, $\mathbf{W}_{\ell,i}^K \in \mathbb{R}^{d \times d_h}$, and $\mathbf{W}_{\ell,i}^V \in \mathbb{R}^{d \times d_h}$ are the projection weight for the $i$-th head in the $\ell$-th layer, where $d_h = d/h$ and $h$ is the number of heads. $\mathbf{H}^{(\ell)}$ denotes the hidden state of the $\ell$-th layer, and $\mathbf{H}^{(0)} = \mathbf{D}$. Each layer takes the hidden state of the previous layer as the query, and the embedding sequence of the source's neighbors as the key and value. The output of the last layer $\mathbf{H}^{(L)}$ will be the edge representations of the source and candidate destinations.

**Prediction.** After the cross-attention module, we encode the elapsed time since last activity of the destination nodes, in order to capture the temporal status of the destinations. The elapsed time is projected to a time-context vector $\mathbb{R}^d$ by a linear layer *TimeProjection* and then concatenated with the edge representation. Finally, we feed them to an MLP to predict the edge likelihood between the source and the destination. Specifically, the predicted score of the $i$-th destination node $v_i$ is computed as:

$$\hat{y}_i = \text{MLP}\left(\text{concat}\left(\mathbf{H}^{(L)}(i), \text{TimeProjection}\left(\Delta t(v_i)\right)\right)\right), \tag{7}$$

where $\Delta t(v_i) = t - t^-(v_i)$ is the elapsed time between the prediction time $t$ and the last activity time of $v_i$, $t^-(v_i)$, and $\mathbf{H}^{(L)}(i)$ is the edge presentation of $(u, v_i, t)$ generated by cross-attention module. The *TimeProjection* function maps the time $t \in \mathbb{R}$ to $\mathbb{R}^d$ by $W_t \cdot t, W_t \in \mathbb{R}^{d \times 1}$. Furthermore, for the scenarios involving excessive seen edges, we encode the repeat time of the candidate edge to capture the inherent repeat patterns in the temporal graph. Here, the repeat time of edge $(a, b)$ refers to the number of times edge $(a, b)$ appears in the dynamic graph upon the prediction time. Similar to the elapsed time encoding, we will first project the repeat time by a linear layer and then concatenate it with the edge representation before the prediction MLP.

**Loss function.** We adopt the Bayesian Personalized Ranking loss [29], which is widely used and well-suited for ranking tasks [19]. Given a source node $u$ interacting with a positive destination node $v_i$ and a negative destination node $v_j$ at time $t$, the loss is defined as: $\mathcal{L}(u, t) = -\log \sigma(\hat{y}_i - \hat{y}_j)$, where $\hat{y}_i$ and $\hat{y}_j$ denote the predicted scores for the positive and negative samples, respectively.

## 4.2 On the Necessity of Learnable Embeddings and Cross-Attention

### 4.2.1 Learnable Node Embeddings Enable Powerful Expressiveness

Most existing methods generate node representations from interaction data, such as edge timestamps and node or edge features. However, in many real-world temporal graphs, including well-established benchmarks [13, 44, 17, 27], these features are often missing or extremely sparse. This is largely because node and edge features are often hard to collect in practice and are not easily aligned across heterogeneous node types [44]. Therefore, many models rely purely on edge timestamps, which significantly limits their ability to distinguish between nodes. In the example shown in Figure 1, existing methods are unable to distinguish nodes $a$ and $f$ because they have identical interaction timestamps and lack unique identifiers.

Recent approaches [5, 21, 38, 46] attempt to mitigate this issue by introducing correlation encodings to capture the relation between the $k$-hop neighborhoods of the source and destination nodes. However, the model's perception is inherently constrained by the choice of $k$. For example, DyGFormer uses neighbor co-occurrence frequency to model correlations between nodes. But in the example shown in Figure 1, it cannot distinguish the candidate edge $(b, a)$ from $(b, f)$, since both $b$ and $a$, and $b$ and $f$ share two common neighbors, i.e. $c$ and $d$, and $d$ and $e$, respectively. It fails to capture that only the co-neighbors of $b$ and $a$ have prior interactions $(c, d, t_0)$, which is a critical signal of a tightly-knit group structure and essential for making accurate predictions. While increasing $k$ may offer broader context, it leads to exponential neighborhood expansion, introducing excessive noise [5] or resulting in memory overhead [44].

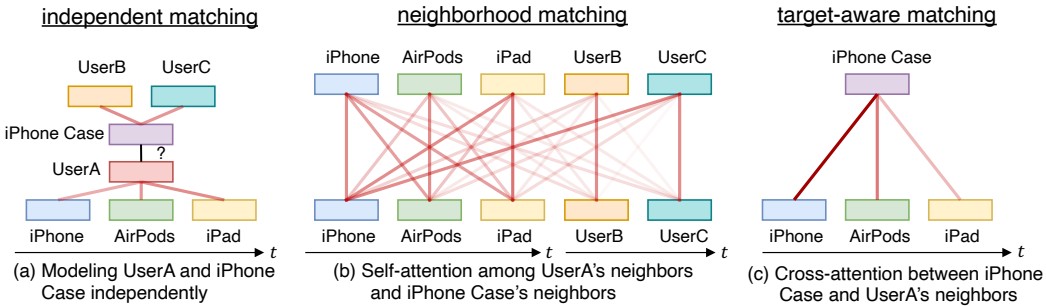

Figure 3: Illustration of different source-destination matching schemes for predicting whether *UserA* is likely to purchase *iPhone Case* at time $t$. Before time $t$, UserA has purchased *iPhone*, *AirPods*, and *iPad* in chronological order, while *UserB* and *UserC* have previously purchased iPhone Case.

These limitations emphasize the necessity of representing nodes with unique identifiers. Similar insights have been observed in the study of the expressiveness of static Graph Neural Networks (GNNs) [4, 20], where the lack of distinctive node embeddings hampers model performance. Notably, Abboud et al. [1] demonstrates that GNNs with randomly initialized node embeddings, which serve as implicit identifiers, can outperform models that rely solely on structural information. Therefore, we introduce learnable embeddings that serve as node identifiers and encode contextual information, enabling the model to distinguish between nodes even in the absence of explicit features.

### 4.2.2    Cross-attention Mechanism Enables Target-aware Matching

Most existing methods [30, 42, 7, 17, 35] do not explicitly model the relationship between source and destination nodes. As illustrated in Figure 1, they typically follow a common paradigm: first, the memory and aggregation modules are used to encode interaction histories into individual node representations; then, an MLP predicts the edge likelihood based on these node representations. This approach models each node's local dynamics independently and neglects the direct matching between node pairs. An exception is DyGFormer [46], which applies self-attention over the combined historical neighbors of both the source and destination nodes, aiming to learn temporal dependencies within and cross the two sets together. While it introduces a degree of cross-node awareness, it focuses on the relation between the source and destination neighborhoods and is not explicitly conditioned on the candidate destination. As a result, when the source and destination neighborhoods lack semantic overlap or are heterogeneous in type, the attention signal becomes diluted or misaligned, leading to weak modeling of compatibility between the destination and the source context.

To better align with the objective of future link prediction, which involves assessing how well a target node fits the source context, we implement target-aware matching through a simple yet effective design: cross-attention between the destination and the source's recent neighbors. This allows each candidate destination to directly attend to the source node's historical interactions and enables the model to capture fine-grained behavioral patterns as well as semantic alignment across entities. Consider the example in Figure 3, which demonstrates how different matching schemes perform in predicting whether *UserA* is likely to purchase *iPhone Case* at time $t$. Subfigure (a) illustrates the independent matching mechanism adopted by most existing methods, where the source (UserA) and the destination (iPhone Case) are modeled independently with their respective neighbors. This hinders the model from recognizing the strong connection between iPhone and iPhone case. In (b), the neighborhood matching mechanism adopted by DyGFormer includes both source and destination neighborhoods, but treats all nodes uniformly. If UserB or UserC has purchased many unrelated items besides iPhone Case, the semantic connection between iPhone and iPhone Case may be overwhelmed. In contrast, our target-aware matching mechanism, shown in (c), enables the candidate iPhone Case to attend directly to iPhone, effectively capturing their semantic relationship. We empirically compare these matching strategies in Section 5.2, demonstrating the advantages of our cross-attention design.

### 4.3    Time Complexity Analysis

We analyze the time complexity of performing future link predictions for a batch of source nodes and their corresponding candidate destinations. Let $B$ denote the batch size, $k$ the number of historical

neighbors considered per node, $q$ the number of candidate destinations per source, $d$ the embedding or feature dimension, and $\bar{d}$ the average node degree in the temporal graph. We compare CRAFT with four state-of-the-art methods: TGAT [42], TGN [30], GraphMixer [7], and DyGFormer [46]. For TGN and TGAT, we follow common practice and consider their standard configurations: TGN with one graph attention layer and TGAT with two. The total time complexity generally consists of two dominant parts: 1) extracting historical neighbors and 2) computing edge likelihoods.

For the extraction of historical neighbors, we consider the common strategy of retrieving the $k$ most recent neighbors [43, 30], following the widely adopted implementation in DyGLib [46]. Alternative neighbor extraction implementation are discussed in the appendix. The dominant cost arises from locating the most recent neighbor among all existing neighbors via binary search, taking $O(\log \bar{d})$ per query. Since existing methods typically gather neighbors for both source and destination nodes, this step takes $O(B(1 + q) \log \bar{d})$. TGAT, which aggregates two-hop neighborhoods via two attention layers, incurs $O(B(1 + q)(1 + k) \log \bar{d})$. In contrast, CRAFT only considers neighbors of the source, reducing the complexity to $O(B \log \bar{d})$. This reduction is particularly beneficial in real-world scenarios, where it is common to evaluate many candidate destinations per source, making efficient neighbor retrieval critical for scalable inference.

To compute edge likelihoods, most existing methods compute node representations by aggregating each node's historical neighbors, followed by an MLP for prediction. The aggregation step dominates computation. TGN, TGAT, GraphMixer, and DyGFormer employ one graph attention layer, two graph attention layers, an MLP-Mixer, and a Transformer encoder, respectively, each incurring various computational complexities per node: $O(kd^2)$, $O(k^2d^2)$, $O(kd^2)$, and $O(kd^2 + k^2d)$. In contrast, CRAFT computes edge likelihoods directly using cross-attention between destinations and the source's neighbors, requiring $O(Bqd^2 + Bkd^2)$ for projections and $O(Bqkd)$ for attention per batch. This approach eliminates the need for neighbor feature projections for each node, significantly improving efficiency, especially when $q$ or $k$ is large.

Table 1 summarizes the results, showing that CRAFT offers superior efficiency in both neighbor extraction and edge likelihood computation. These findings highlight the advantage of CRAFT's simple architecture, making it well-suited for large-scale dynamic systems in practical applications.

## 5  Experiments

In this section, we investigate the effectiveness and efficiency of CRAFT on future link prediction. To fully evaluate the performance of CRAFT on predicting seen and unseen edges, we conduct experiments on 17 datasets, including the TGB-Seq benchmark [44] (ML-20M, Taobao, Yelp, GoogleLocal, Flickr, YouTube, WikiLink), the TGB benchmark [13] (tgbl-review, tgbl-comment, tgbl-coin, tgbl-flights), and six commonly used datasets (wikipedia, reddit, mooc, lastfm, uci, Flights). We compare CRAFT with seven state-of-the-art continuous-time temporal graph models, including JODIE [17], DyRep [35], TGAT [42], TGN [30], CAWN [38], GraphMixer [7], and DyGFormer [46]. The details of datasets and baselines are provided in the appendix.

**Negative destination sampling.** We adopt the negative destination samples from the benchmarks or randomly sample 100 negatives from all potential destinations. Specifically, for datasets from the TGB and TGB-Seq benchmarks, negative samples are provided by the benchmarks themselves: The TGB benchmark samples 100 or 20 negative destinations for each source node from historical and random negative edges [13], while the TGB-Seq benchmark randomly samples 100 negatives for each source [44]. For other datasets without publicly available negatives, we follow the approach outlined in TGB-Seq to randomly select 100 negative destinations for each source node. Importantly,

Table 1: The big-O notation of the time complexity of different temporal graph learning methods, where only the dominant term is retained for clarity.

| Methods | Extract historical neighbors | Compute edge likelihoods |
|---|---|---|
| TGAT [42] | $B(1 + q)(1 + k) \log \bar{d}$ | $B(1 + q)k^2d^2$ |
| TGN [30] | $B(1 + q) \log \bar{d}$ | $B(1 + q)kd^2$ |
| GraphMixer [7] | $B(1 + q) \log \bar{d}$ | $B(1 + q)kd^2$ |
| DyGFormer [46] | $B(1 + q) \log \bar{d}$ | $B(1 + q)(kd^2 + k^2d)$ |
| CRAFT | $B \log \bar{d}$ | $Bqd^2 + Bkd^2 + Bqkd$ |

Table 2: MRR scores (%) of CRAFT and baselines on unseen-dominant datasets. The first, second, and third place rankings are highlighted, accordingly.

| Datasets | ML-20M | Taobao | Yelp | GoogleLocal | Flickr | YouTube | WikiLink | tgbl-review | tgbl-comment |
|---|---|---|---|---|---|---|---|---|---|
| JODIE | $21.16_{\pm0.73}$ | $48.36_{\pm2.18}$ | $69.88_{\pm0.31}$ | $41.86_{\pm1.49}$ | $46.21_{\pm0.83}$ | $41.67_{\pm2.86}$ | $57.94_{\pm1.33}$ | $41.43_{\pm0.15}$ | - |
| DyRep | $19.00_{\pm1.69}$ | $40.03_{\pm2.40}$ | $57.69_{\pm1.05}$ | $37.73_{\pm1.34}$ | $38.04_{\pm4.19}$ | $35.12_{\pm4.13}$ | $42.63_{\pm1.33}$ | $40.06_{\pm0.59}$ | $28.90_{\pm3.30}$ |
| TGAT | $10.47_{\pm0.20}$ | - | - | $19.78_{\pm0.24}$ | $23.53_{\pm3.35}$ | $43.56_{\pm2.53}$ | - | $19.64_{\pm0.23}$ | $56.20_{\pm2.11}$ |
| TGN | $23.99_{\pm0.20}$ | $60.28_{\pm0.54}$ | $69.79_{\pm0.24}$ | $54.13_{\pm1.97}$ | $46.03_{\pm6.78}$ | $55.16_{\pm5.89}$ | $62.94_{\pm2.16}$ | $37.48_{\pm0.23}$ | $37.90_{\pm2.10}$ |
| CAWN | $12.31_{\pm0.02}$ | - | $25.71_{\pm0.09}$ | $18.26_{\pm0.02}$ | $48.69_{\pm6.08}$ | $47.55_{\pm1.08}$ | - | $19.30_{\pm0.10}$ | - |
| GraphMixer | $21.97_{\pm0.17}$ | $31.54_{\pm0.02}$ | $33.96_{\pm0.19}$ | $21.31_{\pm0.14}$ | $45.01_{\pm0.08}$ | $58.87_{\pm0.12}$ | $48.57_{\pm0.02}$ | $36.89_{\pm1.50}$ | $76.17_{\pm0.17}$ |
| DyGFormer | - | - | $21.68_{\pm0.20}$ | $18.39_{\pm0.02}$ | $49.58_{\pm2.87}$ | $46.08_{\pm3.44}$ | - | $22.39_{\pm1.52}$ | $67.03_{\pm0.14}$ |
| SGNN-HN | $33.12_{\pm0.01}$ | $68.58_{\pm0.21}$ | $69.34_{\pm0.44}$ | $62.88_{\pm0.51}$ | $60.15_{\pm0.20}$ | $59.64_{\pm0.22}$ | $69.37_{\pm0.39}$ | $35.62_{\pm0.53}$ | $55.24_{\pm0.22}$ |
| CRAFT | $35.91_{\pm0.65}$ | $70.68_{\pm0.42}$ | $72.69_{\pm0.61}$ | $62.35_{\pm0.46}$ | $62.34_{\pm0.84}$ | $58.92_{\pm0.16}$ | $75.48_{\pm0.84}$ | $41.77_{\pm0.06}$ | $91.72_{\pm0.59}$ |
| Rel.Imprv. | 49.69% | 17.25% | 4.02% | 15.19% | 25.74% | 0.08% | 19.92% | 0.82% | 20.41% |
| Abs.Imprv. | 11.92 | 10.40 | 2.81 | 8.22 | 12.76 | 0.05 | 12.54 | 0.34 | 15.55 |

Table 3: MRR scores (%) of CRAFT-R and baselines on seen-dominant datasets. The first, second, and third place rankings are highlighted accordingly.

| Datasets | wikipedia | reddit | mooc | lastfm | uci | Flights | tgbl-coin | tgbl-flights |
|---|---|---|---|---|---|---|---|---|
| JODIE | $76.48_{\pm1.72}$ | $77.16_{\pm1.27}$ | $19.91_{\pm3.06}$ | $18.49_{\pm2.87}$ | $51.43_{\pm1.74}$ | $20.37_{\pm4.18}$ | - | - |
| DyRep | $67.42_{\pm4.21}$ | $72.33_{\pm2.14}$ | $17.71_{\pm1.27}$ | $17.76_{\pm6.56}$ | $14.00_{\pm2.71}$ | $15.62_{\pm0.47}$ | $45.20_{\pm4.60}$ | $55.60_{\pm1.40}$ |
| TGAT | $72.72_{\pm1.50}$ | $78.03_{\pm0.25}$ | $31.55_{\pm4.14}$ | $24.10_{\pm1.26}$ | $31.50_{\pm0.48}$ | $53.90_{\pm1.07}$ | $60.92_{\pm0.57}$ | - |
| TGN | $82.22_{\pm0.39}$ | $79.25_{\pm0.24}$ | $39.03_{\pm4.53}$ | $25.59_{\pm4.43}$ | $45.29_{\pm6.43}$ | $64.24_{\pm2.11}$ | $58.60_{\pm3.70}$ | $70.50_{\pm2.00}$ |
| CAWN | $86.07_{\pm0.08}$ | $87.66_{\pm0.04}$ | $27.02_{\pm0.57}$ | $35.10_{\pm0.34}$ | $66.96_{\pm0.48}$ | $80.06_{\pm2.10}$ | - | - |
| GraphMixer | $73.57_{\pm1.48}$ | $70.87_{\pm0.38}$ | $29.02_{\pm0.28}$ | $26.76_{\pm1.42}$ | $59.58_{\pm1.00}$ | $42.26_{\pm0.49}$ | $75.57_{\pm0.27}$ | - |
| DyGFormer | $88.69_{\pm0.04}$ | $88.70_{\pm0.05}$ | $42.20_{\pm1.31}$ | $46.50_{\pm0.51}$ | $76.61_{\pm0.26}$ | $84.13_{\pm3.46}$ | $75.17_{\pm0.38}$ | - |
| SGNN-HN | $82.22_{\pm0.35}$ | $87.42_{\pm0.04}$ | $57.25_{\pm0.17}$ | $40.72_{\pm0.26}$ | $57.23_{\pm0.39}$ | $80.80_{\pm0.14}$ | $78.46_{\pm0.40}$ | $89.03_{\pm0.03}$ |
| CRAFT-R | $88.25_{\pm0.26}$ | $89.33_{\pm0.11}$ | $62.32_{\pm0.39}$ | $55.21_{\pm0.23}$ | $75.11_{\pm0.09}$ | $83.16_{\pm0.16}$ | $88.47_{\pm0.25}$ | $91.39_{\pm0.03}$ |
| Rel.Imprv. | -0.50% | 0.71% | 47.68% | 18.73% | -1.96% | -1.15% | 17.07% | 29.63% |
| Abs.Imprv. | -0.44 | 0.63 | 20.12 | 8.71 | -1.50 | -0.97 | 12.90 | 20.89 |

we perform collision checks to ensure that no positive edges are sampled as negative edges for small datasets, which may lead to different results compared to prior works.

**Experimental Settings.** We follow the evaluation protocol of TGB-Seq and TGB benchmarks for all datasets. We use MRR as the evaluation metric, follow the original split of datasets, or chronologically split the datasets into training, validation, and test sets with 75%, 15%, and 15% of the edges, respectively. We follow the configurations in [44, 13, 46] to use the same batch size (200 or 400, accordingly) and learning rate (1e-4) across all methods. The baseline implementation is based on the DyGLib library [46]. For fair comparison, the average results of 3 runs are reported. Other implementation details are provided in the appendix.

### 5.1 Effectiveness and Efficiency of CRAFT on Future Link Prediction

**Experimental Setup.** We categorize the datasets into two groups based on their seen edge ratio. For datasets with a high ratio of seen edges (referred to as *seen-dominant datasets*), we apply repeat time encoding, and denote the variant as CRAFT-R. For datasets with minimal seen edges (referred to as *unseen-dominant datasets*), we use the default CRAFT model without repeat time encoding. The results for all baselines on the TGB-Seq and TGB datasets are directly taken from previous studies [44, 45]. Missing entries indicate cases where the model failed due to either out-of-time (unable to complete a single training epoch within 24 hours) or out-of-memory on a 32GB GPU.

**Effectiveness Evaluation.** Table 2 and Table 3 present the MRR scores on unseen-dominant and seen-dominant datasets, respectively, with *Rel.Imprv.* and *Abs.Imprv.* indicating the relative and absolute improvements over the second-best baseline. Overall, CRAFT achieves the best results on a majority of benchmarks, with over 10% relative improvements on 10 out of 17 datasets compared to the second-best baseline. These results confirm the effectiveness of our architecture and highlight the value of its two key components: unique node identifiers and target-aware matching, which is opposed to the traditional memory and aggregation modules in the baselines. Specifically, CRAFT exhibits superior performance on unseen-dominant datasets, demonstrating that it successfully addresses a key limitation of prior models, which often fail on unseen edge prediction. For seen-dominant datasets, CRAFT-R still performs best on most benchmarks, with only minor underperformance (less than 2%) compared to DyGFormer on three datasets. The reason may be attributed to DyGFormer's use of additional heuristics, such as neighbor co-occurrence frequency, which could provide additional gains in these datasets. However, such design increases computational complexity and lead to out-of-time

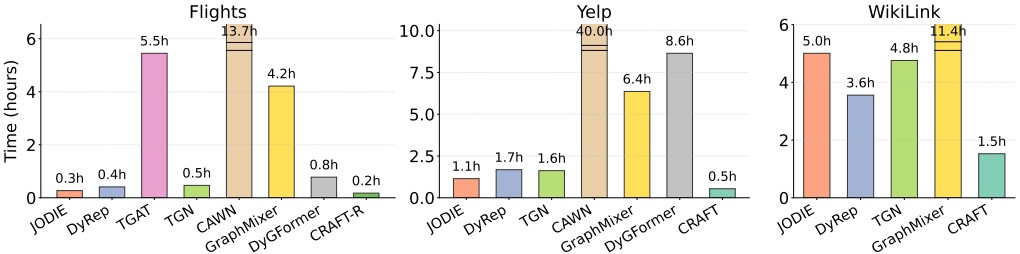

Figure 4: Test set inference time of CRAFT and baselines on Flights, Yelp and WikiLink.

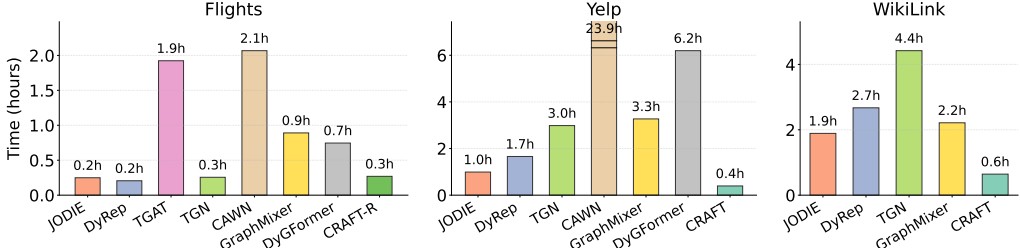

Figure 5: Training time per epoch of CRAFT and baselines on Flights, Yelp and WikiLink.

or out-of-memory failures on large-scale datasets such as WikiLink and tgbl-flights. In contrast, CRAFT maintains high performance with a simpler and more generalizable architecture.

**Efficiency Evaluation.** We evaluate the efficiency of CRAFT by measuring both inference and training times on three representative datasets of increasing scale: Flights (1M edges), Yelp (19M edges), and WikiLink (34M edges). Figure 4 presents the inference time for CRAFT and baseline methods, showing that CRAFT completes prediction on all test edges (each with 100 negative samples) significantly faster than all baselines. Notably, on the larger datasets Yelp and WikiLink, CRAFT requires less than half the time of the most efficient baseline, JODIE. This highlights CRAFT's practical suitability for real-world deployment, where temporal graphs are large and fast inference is essential. Figure 5 illustrates the training time per epoch of CRAFT and the baselines. CRAFT is the most efficient method overall, with the exception of being slightly slower than JODIE and DyRep on the smallest dataset, Flights. However, as the dataset size increases, the efficiency advantage of CRAFT becomes more pronounced. On the WikiLink dataset, many temporal graph learning methods fail to complete a single training epoch within 24 hours, whereas CRAFT finishes one epoch in just 0.6 hours. Compared to the most efficient baseline, JODIE, CRAFT is 1.5× faster on the WikiLink dataset.

## 5.2 Effect of Learnable Node Embeddings and Cross-attention Mechanism

**Experimental Setup.** To verify that unique node identifiers and target-aware matching are critical for future link prediction, we incorporate our designed learnable node embeddings into existing methods, to explore whether it can results in performance improvements. We choose TGAT and DyGFormer as representative methods because they also rely on attention mechanisms. The only difference is the matching mechanisms, where TGAT adopts independent matching, while DyGFormer adopts neighborhood matching. After replacing the original node features with learnable node embeddings, we denotes the modified variants as *TGAT-LE* and *DyGFormer-LE*. For fair comparison, we also use learnable positional encodings for the source and destination neighborhoods in DyGFormer-LE, which is consistent with CRAFT, and we remove the neighbor co-occurrence frequency encoding for DyGFormer-LE. Detailed implementations are provided in the appendix. We evaluate all methods on three large and challenging datasets with minimal seen edges, GoogleLocal, Yelp, and tgbl-review.

**Results.** As shown in Figure 6, comparing each method with its LE variant, we observe significant performance improvements, particularly for TGAT-LE, which achieves up to fivefold gains on Yelp. These results show that learnable node embeddings improve the expressive power of these models and are essential for generalizing to unseen edges in real-world applications. Despite these improvements, both TGAT-LE and DyGFormer-LE still underperform our CRAFT. The performance gap indicates

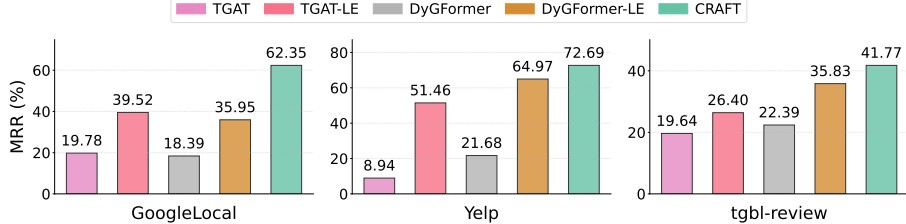

Figure 6: Evaluating the impact of learnable node embeddings and cross-attention: CRAFT vs. TGAT/DyGFormer with and without learnable embeddings (LE).

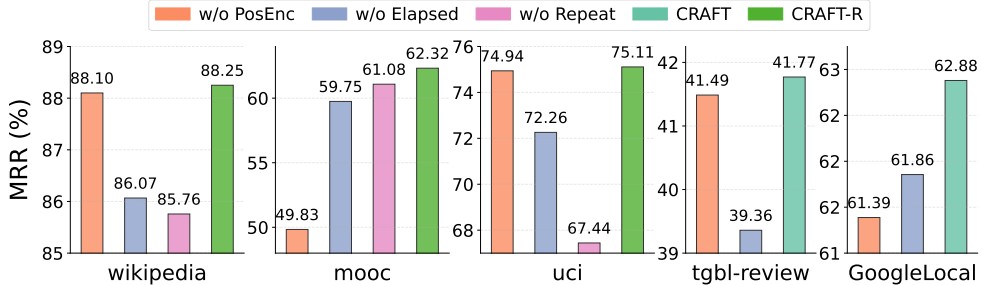

Figure 7: Evaluating the impact of positional encoding, elapsed time encoding, repeat time encoding.

again the importance of target-aware matching, which enables the model to directly assess whether the destination node is compatible with the source context. These findings confirm that the two crucial components in our CRAFT, learnable node embeddings and cross-attention-based target-aware matching, are essential for future link prediction.

### 5.3 Ablation Study: Effect of Positional, Elapsed Time, and Repeat Time Encoding

**Experimental Setup.** We conduct ablation studies to evaluate the impact of three encoding components in CRAFT: positional encoding, elapsed time encoding, and repeat time encoding. For each study, we remove one component while keeping the others intact, and denote the resulting variants as *w/o PosEnc*, *w/o Elapsed*, and *w/o Repeat*, respectively. We evaluate these variants on five representative datasets: wikipedia, mooc, uci, tgbl-review, and GoogleLocal.

**Results.** We find that all three encodings can positively impact the performance of CRAFT. Among them, the elapsed time encoding contributes significantly across all datasets, as it effectively captures the temporal status of each destination node. Positional encoding is especially critical on mooc and GoogleLocal, demonstrating its ability to capture fine-grained temporal order and enable the model to effectively learn from sequential dynamics in these domains. Repeat time encoding plays a prominent role on wikipedia and uci, which exhibit a high proportion of seen edges. While mooc also contains many seen edges, the gain from repeat time encoding is relatively small, possibly because learnable node embeddings and target-aware matching already capture the relationships between the source and destination in this case. Overall, these results confirm the necessity of all three encodings in enabling CRAFT to handle a broad range of temporal dynamics effectively.

## 6  Conclusion

In this paper, we revisit the architectural foundations of temporal graph learning and identify two essential modeling requirements for future link prediction: representing nodes with unique identifiers and performing target-aware matching between the source and destination nodes, which are overlooked in existing methods. Motivated by these observations, we proposed CRAFT, a simple yet effective framework that replaces memory and aggregation modules with learnable node embeddings and a cross-attention mechanism. This design equips CRAFT with strong expressive power and the capacity to explicitly capture destination-conditioned relevance, allowing it to generalize effectively to both seen and unseen edge prediction. Extensive evaluations demonstrate the superiority of CRAFT, establishing it as a practical and scalable solution for real-world future link prediction tasks.

## Acknowledgements

The work was partially done at Gaoling School of Artificial Intelligence, Beijing Key Laboratory of Research on Large Models and Intelligent Governance, Engineering Research Center of Next-Generation Intelligent Search and Recommendation, MOE, and Pazhou Laboratory (Huangpu), Guangzhou, Guangdong 510555, China. This research was supported in part by National Natural Science Foundation of China (No. U2241212, No. 92470128), by National Science and Technology Major Project (2022ZD0114802), by Beijing Outstanding Young Scientist Program No.BJJWZYJH012019100020098 and Huawei-Renmin University joint program on Information Retrieval. We also wish to acknowledge the support provided by the fund for building world-class universities (disciplines) of Renmin University of China, by Engineering Research Center of Next-Generation Intelligent Search and Recommendation, Ministry of Education, by Intelligent Social Governance Interdisciplinary Platform, Major Innovation & Planning Interdisciplinary Platform for the "Double-First Class" Initiative, Public Policy and Decision-making Research Lab, and Public Computing Cloud, Renmin University of China.

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

# A Notation Table

Table 4: Summary of mathematical notations used in the paper.

| Symbol | Description |
|---|---|
| $G = (V, E)$ | Temporal graph with node set $V$ and edge sequence $E$ |
| $V$ | Set of nodes in the temporal graph |
| $E$ | Sequence of edges ordered by non-decreasing timestamps |
| $u$ (or $u_i, u_j$) | Source node in a potential interaction |
| $v$ (or $v_i, v_j$) | Destination node in a potential interaction |
| $e_i = (u_i, v_i, t_i)$ | An edge denoting interaction from source $s_i$ to destination $d_i$ at time $t_i$ |
| $k$ | Number of recent neighbors considered for each source node in CRAFT |
| $h$ | Number of attention heads |
| $d_h$ | Dimension per attention head, where $d_h = d/h$ |
| $\Delta t(v)$ | Elapsed time since last activity of destination node $v$ |
| $t^-(v)$ | Last activity time of destination node $v$ |
| $B$ | Batch size |
| $q$ | Number of candidate destinations per source |
| $d$ | Embedding or feature dimension |
| $\bar{d}$ | Average node degree in the temporal graph |

# B Time Complexity Analysis

## B.1 Other strategies for extracting historical neighbors

An alternative approach for extracting historical neighbors is to maintain a fixed-size window of $k$ most recent neighbors, as implemented in TGB [13]. This strategy necessitates updating the window by removing the oldest neighbor and adding the new one, incurring a cost of $O(B \log B)$ per batch for any temporal graph learning method [13]. For existing methods except CRAFT, this approach is more efficient than maintaining all historical neighbors (Section 4.3) when $(1 + q) \log \bar{d} < \log B$. However, this window-based strategy presents two significant challenges: 1) It requires processing edges in strict chronological order, preventing edge shuffling for sequence-based methods, potentially leading to performance degradation. (We shuffle the training set during CRAFT's training process to learn more robust embeddings for CRAFT.) 2) For datasets with multiple edges occurring simultaneously, this strategy may be infeasible, as the neighbor window could include interactions at the prediction time, rather than before the prediction time, resulting in data leakage.

# C Future Link Prediction and Recommendation

Recommendation is often considered a special application scenario for future link prediction, as stated in works like JODIE [17] and TGN [30]. This is also why many existing datasets used in temporal graph learning come from recommendation systems, such as lastfm (from JODIE for music recommendation) and tgbl-review (from TGB [13], a dataset for e-commerce recommendation curated from Amazon.com). Temporal graph learning covers a wider range of application scenarios, from education (the mooc dataset from JODIE), to transportation (tgbl-flight [13]), and transactions (tgbl-coin [13]). In essence, temporal graph learning aims to capture a variety of evolving patterns in real-world dynamic systems, while recommendation is one specific use case. Recommendation focuses on bipartite graphs (user-item interactions), while temporal graph learning aims to work on general temporal graphs, including both bipartite and non-bipartite graphs.

To further assess the effectiveness of CRAFT, we compare it with two widely recognized recommendation methods: SASRec [14] and SGNN-HN [25], on bipartite datasets. Figures 8 and 9 display the MRR scores of CRAFT (and CRAFT-R), SASRec, and SGNN-HN on four small bipartite datasets and five large bipartite datasets, respectively. We observe that CRAFT achieves the best performance on most datasets, except for the GoogleLocal dataset, where it slightly trails behind SGNN-HN. Generally, CRAFT-R outperforms both SASRec and SGNN-HN on seen-dominant datasets with a larger margin, compared to the results on unseen-dominant datasets. These results suggest that existing

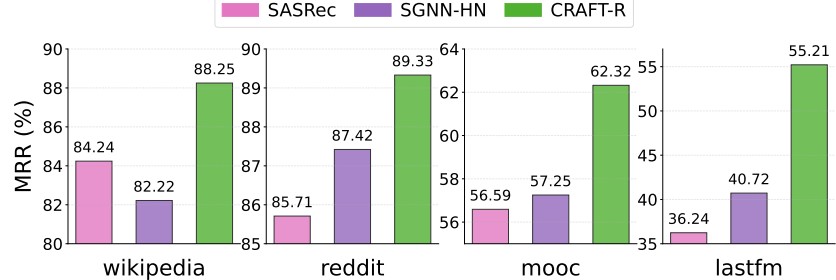

Figure 8: MRR scores of CRAFT-R, SASRec and SGNN-HN on seen-dominant bipartite datasets.

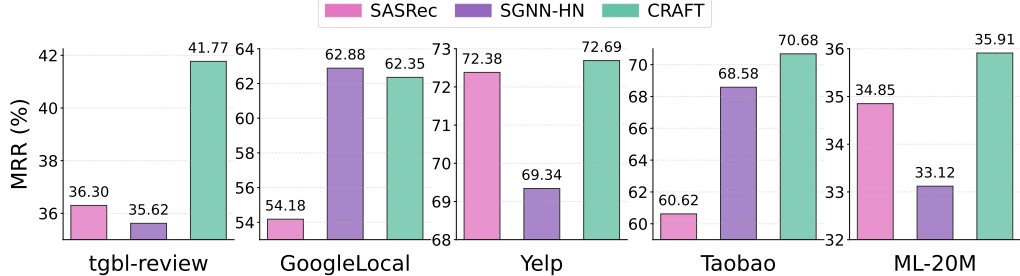

Figure 9: MRR scores of CRAFT, SASRec and SGNN-HN on unseen-dominant bipartite datasets.

recommendation methods perform worse than temporal graph learning methods in seen-dominant scenarios, while our proposed CRAFT achieves the best of both worlds.

# D  Experiment Details and Additional Empirical Study

## D.1  Code Availability

The implementation of CRAFT and baselines are based on DyGLib [46]. The implementation of CRAFT and all commands to reproduce the experimental results are publicly available at `https://github.com/luyi256/CRAFT`.

## D.2  Datasets

In our experiments, we adopt a diverse set of 17 datasets: seven from TGB-Seq [44], four from TGB [13], and six commonly used datasets in the field. Table 5 presents the key statistics of these datasets for reference. Detailed descriptions of the datasets are available in their respective original papers: TGB-Seq datasets in [44], TGB datasets in [13], wikipedia, reddit, mooc, lastfm in [17], uci in [26], Flights in [27]. While other datasets have been proposed in the literature [27], we exclude those with an insufficient number of nodes to ensure robust evaluation.

## D.3  Baselines

We provide a concise overview of the baselines used in our experiments, focusing on their memory and aggregation modules.

**JODIE** incorporates both memory and aggregation modules. The memory module utilizes an RNN to encode historical interactions into a compressed state. The aggregation (embedding) module generates current node representations by projecting the memory states with a time-context vector.

**DyRep** employs an RNN for memory state updates, similar to JODIE. It leverages temporal graph attention to transform interaction history, using these transformed representations as input for the memory module. The aggregation module is a simple identity mapping, directly outputting nodes' memory states.

Table 5: Datasets statistics: seven TGB-Seq datasets, four TGB datasets, and six commonly used datasets. This table is partially adopted from TGB-Seq [44].

| Dataset | Nodes (users/items) | Edges | Timestamps | Repeat ratio(%) | Bipartite | Domain |
|---|---|---|---|---|---|---|
| ML-20M | 100,785/9,646 | 14,494,325 | 9,993,250 | 0 | ✓ | Movie rating |
| Taobao | 760,617/863,016 | 18,853,792 | 139,171 | 16.58 | ✓ | E-commerce interaction |
| Yelp | 1,338,688/405,081 | 19,760,293 | 14,646,734 | 25.18 | ✓ | Business review |
| GoogleLocal | 206,244/267,336 | 1,913,967 | 1,771,060 | 0 | ✓ | Business review |
| Flickr | 233,836 | 7,223,559 | 134 | 0 | × | Who-To-Follow |
| YouTube | 402,422 | 3,288,028 | 203 | 0 | × | Who-To-Follow |
| WikiLink | 1,361,972 | 34,163,774 | 2,198 | 0 | × | Web link |
| tgbl-review | 352,636/298,590 | 4,873,540 | 6,865 | 0.19 | ✓ | E-commerce review |
| tgbl-coin | 638,486 | 22,809,486 | 1,295,720 | 82.93 | × | Transaction |
| tgbl-comment | 994,790 | 44,314,507 | 30,998,030 | 19.81 | × | Reply network |
| tgbl-flight | 18,143 | 67,169,570 | 1,385 | 96.48 | × | Transport |
| Wikipedia | 8,227/1,000 | 157,474 | 152,757 | 88.41 | ✓ | Interaction |
| Reddit | 10,000/984 | 672,447 | 669,065 | 88.32 | ✓ | Reply network |
| MOOC | 7,047/97 | 411,749 | 345,600 | 56.66 | ✓ | Interaction |
| LastFM | 980/1,000 | 1,293,103 | 1,283,614 | 88.01 | ✓ | Interaction |
| UCI | 1,899 | 59,835 | 58,911 | 66.06 | × | Social contact |
| Flights | 13,169 | 1,927,145 | 122 | 79.50 | × | Transport |

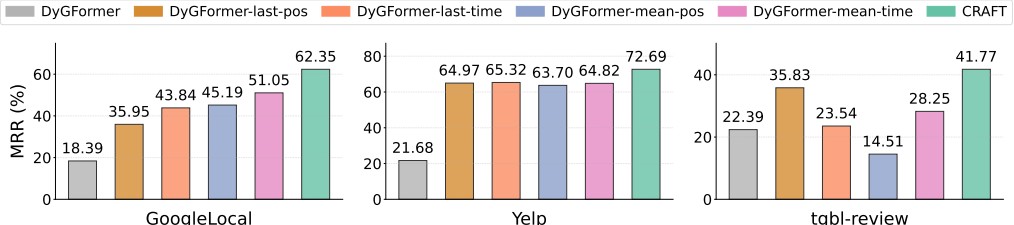

Figure 10: Evaluating the impact of learnable node embeddings and cross-attention: CRAFT and more variants of DyGFormer.

**TGN** uses a GRU for memory state updates and temporal graph attention to aggregate historical neighbor information for node embeddings. This information encompasses neighbors' memory states, features, edge features, and timestamps.

**TGAT** pioneered temporal graph attention, employing multiple layers to aggregate historical neighbors without a memory module.

**CAWN** extracts temporal random walks for each node and applies an anonymization strategy to ensure inductive learning. These causal anonymous walks are then encoded by an RNN to generate node embeddings.

**GraphMixer** is a sequence-based method that simply aggregates historical neighbor information using MLP-Mixer without a memory module. It introduces a fixed time encoding scheme, demonstrating superior effectiveness compared to trainable time encoding.

**DyGFormer**, another sequence-based method, utilizes a transformer encoder in its aggregation module. It computes neighbor co-occurrence frequency between source and destination nodes to capture their relationship in terms of common neighbors.

### D.4 Collision Check for Small Datasets

For datasets with a small number of nodes (wikipedia, reddit, mooc, lastfm, uci and Flights), we implement a collision check throughout the entire training process, including training, validation, and testing phases. This approach enhances the performance of most temporal graph learning methods compared to the results of the wikipedia and reddit datasets reported in [44]. This improvement is attributed to the elimination of false positive samples, which prevents model confusion during training and validation.

Table 6: AP scores (%) of CRAFT and baselines on four commonly-adopted datasets.

| Dataset | CRAFT | DyGFormer | GraphMixer | TGN | DyRep | JODIE | TGAT | CAWN |
|---|---|---|---|---|---|---|---|---|
| wikipedia | 98.07±0.10 | 99.08±0.07 | 97.28±0.25 | 98.60±0.10 | 95.18±0.93 | 96.94±0.30 | 97.32±0.09 | 98.80±0.02 |
| reddit | 99.32±0.04 | 99.27±0.05 | 97.50±0.05 | 98.71±0.01 | 98.28±0.06 | 98.50±0.08 | 98.49±0.00 | 99.15±0.01 |
| lastfm | 93.93±0.03 | 92.99±0.20 | 75.99±0.22 | 78.24±2.39 | 70.43±3.42 | 69.23±0.75 | 73.00±0.35 | 86.91±0.09 |
| mooc | 96.15±0.01 | 89.36±0.41 | 83.51±0.17 | 91.00±3.75 | 81.30±1.87 | 82.18±1.37 | 86.34±0.49 | 81.24±0.44 |

### D.5 Implementation of TGAT-LE and DyGFormer-LE

In Section 5.2, we introduce TGAT-LE (TGAT with learnable embeddings) and DyGFormer-LE (DyGFormer with learnable embeddings) to demonstrate the effectiveness of learnable embeddings and the cross-attention mechanism. We provide detailed implementations of these variants below.

**TGAT-LE.** We replace node features with learnable embeddings without further modifications. While the standard TGAT configuration uses two layers of temporal graph attention, TGAT-LE employs only one layer. Due to the out-of-time issue, TGAT cannot run on the Yelp dataset with two layers [44]. Therefore, we evaluate TGAT with one layer for Yelp, while other TGAT results are directly adopted from Table 2 in the main paper. Notably, TGAT-LE outperforms TGAT even with a single layer, demonstrating the strong expressiveness of learnable embeddings.

**DyGFormer-LE.** In this variant, we replace the original node features with learnable node embeddings and remove the neighbor co-occurrence frequency encoding scheme. Additionally, we replace the time encoding with learnable positional encoding, similar to the approach used in CRAFT. After passing through the transformer encoder, we use the output for the last neighbor of both the source and destination nodes as their respective node representations, following a strategy commonly used in Transformer-based sequence models [14]. To better differentiate this variant, we refer to it as *DyGFormer-last-pos*, as it uses the output of the last neighbor from the transformer encoder and adopts positional encoding.

**More Variants of DyGFormer.** An alternative strategy for producing node representations is to use *mean-pooling* over the outputs of all neighbors from the transformer encoder. Additionally, temporal information captured by positional encoding can, in principle, also be conveyed through time encoding. We explore these options and introduce three variants of DyGFormer: 1) *DyGFormer-last-time*, which uses the last neighbor output with time encoding instead of positional encoding; 2) *DyGFormer-mean-pos*, which uses mean-pooling with positional encoding; and 3) *DyGFormer-mean-time*, which uses mean-pooling with time encoding. For time encoding, we concatenate a time-context vector — obtained by projecting the time interval between the prediction time and each historical interaction through a linear layer — with the corresponding node embedding. The results of these variants are shown in Table 10. The performance of these variants varies across different datasets, suggesting that different datasets exhibit varying interaction patterns. For instance, the GoogleLocal dataset benefits from the mean-pooling strategy, possibly due to the consistent preferences of users in this dataset. Nevertheless, we observe that all variants underperform CRAFT, highlighting the superiority of our cross-attention mechanism.

### D.6 Average-Precision Performance Comparison

Recent studies show that MRR is better suited for tasks with multiple negative samples, as it captures the ranking of the positive edge among negatives more effectively. Moreover, many dynamic graph learning benchmarks [13, 44] use MRR as the primary metric for link prediction tasks. Therefore, we chose MRR as our main evaluation metric. For reference, we also include the AP results in Table 6, which align with the experiments in Section 5 and highlight CRAFT's superiority.

### D.7 Empirical Study: Positional Encoding vs. Time Encoding

In CRAFT, we use positional encoding to capture the temporal order of the source's neighbors. An alternative approach is time encoding, which encodes the time interval between the prediction time and the timestamps of historical interactions. We compare the performance of CRAFT using positional encoding and time encoding in Figure 11. For time encoding, we concatenate a time-context vector, which is obtained by projecting the time interval through a linear layer, with the

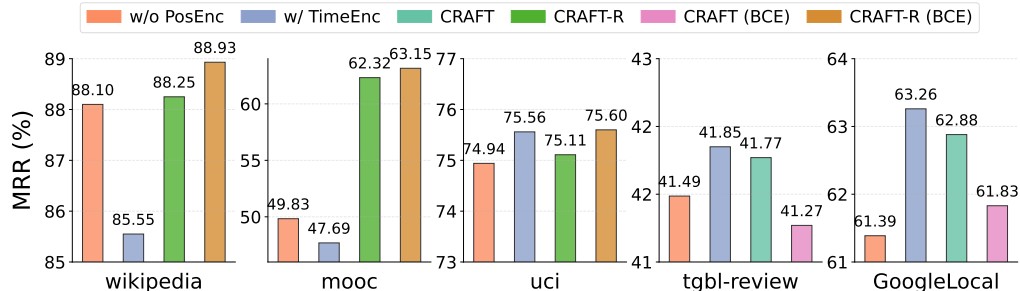

Figure 11: Comparisons of positional encoding vs. time encoding and BPR loss vs. BCE loss.

Table 7: The tuning ranges of hyperparameters for CRAFT.

| Hyperparameter | Tuning Range | Descriptions |
|---|---|---|
| $N_{\text{neighbors}}$ | [30, 60, 90, 120] | the number of neighbors used for each source |
| $p_{\text{hidden\_dropout}}$ | [0.1, 0.2, 0.3, 0.5] | the dropout rate for FFNs and MLPs |
| $p_{\text{attn\_dropout}}$ | [0.1, 0.2] | the dropout rate for attention scores |
| $p_{\text{emb\_dropout}}$ | [0.1, 0.2] | the dropout rate for node embeddings |
| $N_{\text{layers}}$ | [1, 2] | the number of layers in cross-attention |

node embedding of each neighbor of the source. These concatenated vectors serve as the key and value inputs to the cross-attention module. In the figure, *w/o PosEnc* refers to models without either positional or time encoding (same as in Figure 7), *w/ TimeEnc* refers to models using time encoding instead of positional encoding, and the original CRAFT (or CRAFT-R) uses positional encoding. We observe that CRAFT with positional encoding and CRAFT with time encoding perform comparably across most datasets. However, CRAFT with positional encoding significantly outperforms CRAFT with time encoding on the wikipedia and mooc datasets. This suggests that using time encoding introduces excessive noise in these datasets, whereas the temporal order provided by positional encoding is sufficient for capturing the sequential dynamics.

## D.8 Empirical Study: BPR loss vs. BCE loss

Most temporal graph learning methods are trained using the Binary Cross Entropy (BCE) loss, whereas CRAFT adopts the BPR loss. To assess the impact of the loss function, we compare the performance of CRAFT trained with BPR loss and BCE loss in Figure 11. The results show that both versions perform comparably across most datasets. Among the five evaluated datasets, only GoogleLocal exhibits a performance gap greater than 1% between the two losses. These findings highlight the robustness of CRAFT and indicate that its performance is largely unaffected by the choice between BPR and BCE loss functions.

## D.9 Hyperparameters

The hyperparameters for the baselines on the TGB-Seq benchmark, TGB benchmark, and six commonly used datasets (wikipedia, reddit, mooc, lastfm, uci, and Flights) are selected based on the best configurations reported in [44],[13], and[46], respectively. Thanks to CRAFT's simple architecture, it involves only a limited number of hyperparameters. We fix the batch size across all methods, and set the embedding size to 64 for small datasets and 128 for large ones. Table 7 presents the hyperparameters and their tuning ranges. We perform grid search over these ranges using the validation set to determine the optimal settings. The final hyperparameters for CRAFT, along with the batch size and embedding size per dataset, are summarized in Table 8.

Table 8: Optimal hyperparameters and embedding size $F$ for CRAFT, and the fixed batch size $B$ on various datasets.

| Dataset | $N_{\mathrm{neighbors}}$ | $p_{\mathrm{hidden\_dropout}}$ | $p_{\mathrm{attn\_dropout}}$ | $p_{\mathrm{emb\_dropout}}$ | $N_{\mathrm{layers}}$ | $B$ | $F$ |
|---|---|---|---|---|---|---|---|
| uci | 30 | 0.3 | 0.2 | 0.2 | 1 | 200 | 64 |
| wikipedia | 120 | 0.1 | 0.1 | 0.2 | 1 | 200 | 64 |
| reddit | 120 | 0.1 | 0.2 | 0.1 | 1 | 200 | 64 |
| mooc | 30 | 0.1 | 0.1 | 0.1 | 2 | 200 | 64 |
| Flights | 90 | 0.2 | 0.2 | 0.2 | 1 | 200 | 64 |
| lastfm | 120 | 0.1 | 0.1 | 0.1 | 1 | 200 | 64 |
| GoogleLocal | 60 | 0.2 | 0.1 | 0.2 | 2 | 200 | 128 |
| Flickr | 90 | 0.1 | 0.1 | 0.2 | 1 | 400 | 128 |
| YouTube | 90 | 0.2 | 0.2 | 0.2 | 2 | 400 | 128 |
| Taobao | 60 | 0.2 | 0.1 | 0.2 | 2 | 400 | 128 |
| Yelp | 60 | 0.2 | 0.2 | 0.2 | 2 | 400 | 128 |
| ML-20M | 60 | 0.2 | 0.2 | 0.2 | 1 | 400 | 128 |
| Wikilink | 60 | 0.2 | 0.2 | 0.2 | 1 | 400 | 128 |
| tgbl-review | 120 | 0.1 | 0.2 | 0.1 | 1 | 200 | 128 |
| tgbl-coin | 60 | 0.2 | 0.2 | 0.2 | 1 | 200 | 128 |
| tgbl-comment | 60 | 0.2 | 0.1 | 0.2 | 1 | 200 | 128 |
| tgbl-flight | 90 | 0.2 | 0.2 | 0.2 | 1 | 200 | 128 |

# E  Discussion

## E.1  Broader Impact

This paper introduces CRAFT, an efficient and effective temporal graph learning method for future link prediction. CRAFT can be applied across a range of real-world scenarios, including user recommendation on social media, product recommendation on e-commerce platforms, and interaction forecasting in social, financial, and gaming networks. By enabling more accurate and scalable future link prediction, CRAFT has the potential to enhance user experience and improve overall system utility in these domains.

## E.2  Transductive and Inductive Future Link Prediction

The transductive setting for future link prediction focuses on forecasting future links for nodes observed during training, while the inductive setting targets predicting links involving previously unseen nodes [30]. In this paper, we do not explicitly separate the transductive and inductive tasks. All datasets follow the original training, validation, and test splits provided by existing benchmarks or are split chronologically. As a result, our evaluation naturally encompasses both settings.

CRAFT is primarily designed for transductive future link prediction, as it relies on historical interactions to learn meaningful node embeddings. In the inductive setting, however, the source or destination node has little to no interaction history—a situation that frequently arises in real-world dynamic graphs, such as when a new user joins a social network or a new item is introduced to an e-commerce platform. Without prior interactions, it is difficult for any method to infer a new node's behavioral preferences. This scenario aligns with the well-known *cold-start* problem in recommendation systems [31], where common solutions involve incorporating node profile features and estimating similarity with known nodes. We view this inductive scenario as a distinct challenge from standard future link prediction, which focuses on leveraging interaction history to model node preferences and predict future links based on dynamic behavioral patterns. Addressing the inductive case effectively may require additional assumptions or auxiliary data beyond the scope of this work.

Another perspective on inductive learning is to eliminate training costs by expecting the model to generalize directly to unobserved nodes, assuming these nodes have sufficient historical interactions to infer their preferences. However, as demonstrated in Section 5.2, incorporating learnable node embeddings substantially improves the performance of temporal graph learning methods by enhancing their expressive power. Moreover, the experimental results in Table 2 and Table 3 show that CRAFT consistently outperforms baselines designed for inductive learning, despite their more complex architectures. While inductive learning remains a valuable research direction, we argue that learning

Table 9: Comparison of CRAFT-core with features vs. embeddings.

| Datasets | CRAFT-core_w_feat | CRAFT-core_w_embed |
|----------|-------------------|--------------------|
| tgbl-flight | 86.36 | 90.61 |

Table 10: Comparison of the original CRAFT and CRAFT with features.

| Method | wikipedia | reddit | mooc |
|--------|-----------|--------|------|
| CRAFT | 88.25±0.26 | 89.33±0.11 | 62.32±0.39 |
| CRAFT_add_feat | 88.40±0.39 | 89.42±0.04 | 61.03±0.62 |

from historical interactions is a fundamental requirement for future link prediction, which depends on accurately modeling node-specific interaction patterns. Therefore, training CRAFT on all available historical interactions to derive meaningful node embeddings offers a more effective approach.

### E.3  Node and Edge Features

The current version of CRAFT does not incorporate any node or edge features. This decision is motivated by several considerations.

**First**, most benchmark datasets lack such features. This is because node features in real-world dynamic graphs are often incomplete, noisy, and difficult to align across node types, particularly in bipartite and heterogeneous graphs. For example, the GoogleLocal dataset [44], a real-world dataset of restaurant reviews from Google Maps, includes two node types: users and places, whose features (e.g., user profile vs. place attributes like price) cannot be aligned. Furthermore, many features are missing, like the price feature for over 267,000 out of 267,336 places. These issues explain why many temporal graph benchmarks often lack features. If the features are sparse or poor quality, they can introduce noise and degrade performance.

**Second**, existing studies have shown that interaction data provides more valuable information than node features for link prediction, whether in static graphs [23] or temporal graphs [44]. CRAFT, which does not use any features, still achieves superior performance compared to the baselines that incorporate features. This highlights the critical importance of modeling temporal interactions and the limitations of current methods in fully leveraging interaction data.

**Third**, while node features can differentiate nodes, they cannot capture interaction patterns as effectively as learnable node embeddings. To evaluate the contribution of node features to future link prediction in existing benchmarks, we conduct experiments on the tgbl-flight dataset, which is the only dataset among those we considered that provides node features. We compare CRAFT-core_w_embed (which uses learned node embeddings) with CRAFT-core_w_feat (which replaces embeddings with raw node features). Here, CRAFT-core denotes the CRAFT model without the repeated time and last elapsed time encodings, which we remove to isolate their effects and focus on the role of node representations. As noted in TGB [13], the node features for each airport tgbl-flight include attributes like the ISO region code, longitude, and latitude, which can differentiate nodes. However, Table 9 shows that the node embeddings is more effective than node features in CRAFT. This is because node embeddings are learned from the interaction data, which better captures nodes' interaction patterns.

That said, CRAFT is flexible and can easily accommodate features if needed. Node features can be concatenated with learnable node embeddings, while edge features can be appended to the source's neighbor embeddings or combined with positional encodings to indicate edge types. To demonstrate this flexibility, we conduct a simple experiment in which both node and edge features (when available) are first projected through a linear layer and then concatenated with the learnable embeddings. Table 10 shows that CRAFT_add_feat performs comparably to the original CRAFT, suggesting that this method can serve as a basic variant of CRAFT for incorporating features when needed. While CRAFT_add_feat slightly outperforms CRAFT on the wikipedia and reddit datasets, it underperforms on mooc. This reinforces our previous point that node features may often lack high quality or be less valuable for link prediction.

### E.4 Negative Sampling Strategies

Poursafaei et al. [27] proposed two challenging negative sampling strategies for future link prediction: historical negative sampling and inductive negative sampling. In this work, we adopt multiple negative samples per positive instance and follow the random negative sampling strategy used in prior benchmarks [13, 44], as using multiple random negatives per sample is sufficiently challenging for the models and provides a fair basis for comparison.

### E.5 Limitations and Future Work

CRAFT is specifically designed for future link prediction and does not directly support other tasks such as dynamic node classification. Our design focuses on modeling the relationship between the destination node and the source node's interaction history, without maintaining dynamic node representations over time. To extend CRAFT to support dynamic node classification, a straightforward approach is to update node embeddings continuously via gradient descent. This has the potential advantage of enabling a single model to support multiple tasks, eliminating the need for training separate models as required in prior work. Exploring such unified and flexible modeling capabilities is a promising direction for future research.

