# OpenReview forum: "Future Link Prediction Without Memory or Aggregation"
_NeurIPS.cc/2025/Conference — NeurIPS 2025 poster_

### Official Review · Reviewer_657h · 2025-06-27

**Clarity:** 3
**Significance:** 3
**Originality:** 3
**Rating:** 4
**Confidence:** 3

**Summary:**

This paper introduces CRAFT, which discards memory and aggregation modules for future link prediction on temporal graphs.

**Questions:**

Please see weakness

**Ethical Concerns:**

["NO or VERY MINOR ethics concerns only"]

**Final Justification:**

The authors’ responses have addressed all of my concerns. Although I still believe that the embedding based approach, without additional mechanisms, may struggle to handle unseen nodes, overall the manuscript meets the bar for acceptance.

**Limitations:**

Yes

**Quality:**

2

**Strengths And Weaknesses:**

Strength:
1. The paper is well-organized and the overall presentation is clear and coherent.
2. The proposed method achieves strong empirical results on the task considered in the paper.

Weakness:
1. The roles of the embedding and memory components appear to be quite similar in functionality, yet the paper does not clearly differentiate between the two.
2. The link prediction task formulation differs from conventional dynamic graph representation learning settings. It remains uncertain whether the proposed method outperforms existing models such as DyGFormer under a standardized task formulation.
3. In Section 4.3, the number of candidate destination nodes is controlled by a hyperparameter q. The method used to select destination nodes may significantly influence the final results. However, the paper lacks a sensitivity analysis on this hyperparameter.
4. Although the paper claims that the memory component does not help in handling unseen nodes, the embedding-based approach does not seem to offer any specific mechanism for such cases either. This raises the question of whether the two can, in fact, be treated equivalently.
5. The paper lacks a concrete case study to visually or intuitively validate its key contributions.

---

> ### Author Rebuttal · Authors · 2025-07-29
>
> > W1. "The roles of the embedding and memory components appear to be quite similar in functionality, yet the paper does not clearly differentiate between the two."
> >
> > W4. "Although the paper claims that the memory component does not help in handling unseen nodes, the embedding-based approach does not seem to offer any specific mechanism for such cases either. This raises the question of whether the two can, in fact, be treated equivalently."
>
> Thank you for your thoughtful questions. **While both the embedding and memory components share the goal of representing a node's interaction history in a compressed form, they differ significantly in design and capability.**
>
> **Design**: The memory component typically uses an RNN model to update each node's memory vector when new interactions occur. The memory of each node is initialized as a zero vector and updated by the RNN, which takes the memory and new interaction data as inputs. In contrast, node embeddings in CRAFT are learnable, updated through gradient descent, and directly encode the interaction patterns of nodes.
>
> **Capability**: The memory component struggles with unseen edges (as noted in [R3]), as it does not serve as a unique identifier for nodes, unlike learnable node embeddings. Specifically, when two nodes always interact with other nodes at the same time, their memory become indistinguishable because the RNN input for both nodes is identical, especially when no features are available.
>
> Consider this example (inspired by [R3]): In a temporal graph with two disconnected groups of nodes, Group A and Group B, each node in Group A corresponds to a node in Group B, and they interact identically within their groups. For example, $(A_1, A_2, t)$ in Group A corresponds to $(B_1, B_2, t)$ in Group B. The memory component cannot distinguish between corresponding nodes, such as $A_1$ and $B_1$, because the RNN input for both is the same. However, learnable node embeddings can distinguish between any pair of nodes as they serve as unique identifiers.
>
> This lack of distinguishability limits the memory component's expressiveness, preventing it from differentiating nodes with similar interaction patterns. This issue is especially critical in datasets with many unseen edges, where the model needs expressive node representations to rank multiple candidates accurately, rather than relying on memorizing seen edges. **To empirically compare the memory component and learnable node embeddings, we conducted experiments using a simple model `Embedding`**, which only utilizes node embeddings. In this model, the probability of an edge $(s,d)$ is predicted using the dot product of the embeddings of nodes $s$ and $d$. Formally, the score for edge $(s,d)$ is computed as:
>
> $$
> \text{score}(s,d) = \text{e}(s)^T \cdot \text{e}(d), \quad \text{where } \text{e}(s), \text{e}(d) \in \mathbb{R}^d.
> $$
>
> We compared this model with JODIE, which incorporates a memory component and a simple time projection layer as its aggregation module. As shown in Table 1, **Embedding outperforms JODIE by a large margin, demonstrating the significant capability gap between node embeddings and memory components**.
>
> Table 1: Embedding vs. JODIE
> |Dataset|GoogleLocal|YouTube|Flickr|reddit|Flights|
> |-|-|-|-|-|-|
> |Embedding|**44.81**|**49.76**|**55.03**|**81.02**|**79.84**|
> |JODIE|41.86|41.67|46.21|77.16|20.37|
>
> ---
> > W2. "The link prediction task formulation differs from conventional dynamic graph representation learning settings. It remains uncertain whether the proposed method outperforms existing models such as DyGFormer under a standardized task formulation."
>
> There appears to be a misunderstanding. **The task in this paper is exactly the same as in conventional dynamic graph representation learning settings**, such as those in DyGFormer, TGN, JODIE, etc. Formally, the task is to predict the probability of an edge $(s,d)$ occurring at time $t$ based on the interaction history of $s$ and $d$. **For this reason, we can directly adopt many of the reported results of existing methods from the literature, including those on the TGB benchmark, as evaluated in [R1] by the authors of DyGFormer**.
>
> The misunderstanding may arise from our use of MRR as the evaluation metric, instead of AP, as in the original DyGFormer paper. Recent studies show that MRR is better suited for tasks with multiple negative samples, as it captures the ranking of the positive edge among negatives more effectively. Moreover, many dynamic graph learning benchmarks [R2, R3] use MRR as the primary metric for link prediction tasks. Therefore, we chose MRR as our main evaluation metric. For reference, we also include the AP results in Table 2, which align with the experiments in our manuscript and highlight CRAFT's superiority over the baselines.
>
> Table 2: AP results
> |Dataset|CRAFT|DyGFormer|GraphMixer|TGN|DyRep|JODIE|TGAT|CAWN|
> |-|-|-|-|-|-|-|-|-|
> |wikipedia|98.07±0.10|**99.08±0.07**|97.28±0.25|98.60±0.10|95.18±0.93|96.94±0.30|97.32±0.09|98.80±0.02|
> |reddit|**99.32±0.04**|99.27±0.05|97.50±0.05|98.71±0.01|98.28±0.06|98.50±0.08|98.49±0.00|99.15±0.01|
> |lastfm|**93.93±0.03**|92.99±0.20|75.99±0.22|78.24±2.39|70.43±3.42|69.23±0.75|73.00±0.35|86.91±0.09|
> |mooc|**96.15±0.01**|89.36±0.41|83.51±0.17|91.00±3.75|81.30±1.87|82.18±1.37|86.34±0.49|81.24±0.44|
>
> ---
> > W3. "In Section 4.3, the number of candidate destination nodes is controlled by a hyperparameter q. The method used to select destination nodes may significantly influence the final results. However, the paper lacks a sensitivity analysis on this hyperparameter."
>
> Thank you for this insightful point. For datasets from the TGB and TGB-Seq benchmarks, **negative samples are provided by the benchmarks themselves to ensure fairness**: The TGB benchmark samples 100 or 20 negative destinations for each source node from historical and random negative edges [R1], while the TGB-Seq benchmark randomly samples 100 negatives for each source [R2]. For other datasets without publicly available negatives, we follow the approach outlined in [R2] to randomly select 100 negative destinations for each source node.
>
> **Increasing $q$ typically raises the evaluation difficulty and lowers MRR scores**, as methods must rank the positive destination among a larger pool of candidates. To assess the robustness of CRAFT, we conducted experiments with $q=500$ on the lastfm and wikipedia datasets. As shown in Table 3, CRAFT either outperforms or performs comparably to the best baseline when $q=500$. Notably, the significant superiority of CRAFT over the baselines on lastfm with $q=100$ is also maintained with $q=500$, further demonstrating CRAFT's robustness.
>
> We also experimented with CAWN, which performed third-best on wikipedia and lastfm. However, running experiments with CAWN took over 80 hours to complete, which explains why existing benchmarks typically use $q=100$ to balance efficiency and evaluation thoroughness[R2].
>
> Table 3: MRR with $q=500$
> |Dataset|CRAFT|DyGFormer|GraphMixer|
> |-|-|-|-|
> |lastfm|**28.56**|21.20|10.91|
> |wikipedia|69.08|**70.11**|49.98|
>
> ---
> > W5. "The paper lacks a concrete case study to visually or intuitively validate its key contributions."
>
> Thank you for this valuable point. We have provided two case studies in Section 4.2 to demonstrate the necessity of our key components: learnable node embeddings and the cross-attention mechanism. Figure 1 shows that even methods like DyGFormer, which aim to improve the ability to distinguish between nodes, fail in a simple example, highlighting the need for learnable node embeddings to provide unique identifiers. Figure 3 compares CRAFT's cross-attention mechanism with existing methods and explains its advantages.
>
> In response to your feedback, we plan to combine the temporal graph example from Figure 1 with the example in our responses to W1 and W4 into a unified graph. This will further illustrate the limitations of existing methods and the importance of node embeddings. If you have any suggestions, we are happy to discuss further.
>
> [R1] An Empirical Evaluation of Temporal Graph Benchmark. Arxiv'23
>
> [R2] Temporal graph benchmark for machine learning on temporal graphs. NeurIPS'23
>
> [R3] TGB-Seq Benchmark: Challenging Temporal GNNs with Complex Sequential Dynamics. ICLR'25
>
> [R4] Towards Better Evaluation for Dynamic Link Prediction. NeurIPS'22

---

> > ### Comment · Reviewer_657h · 2025-08-05
> >
> > Thank you for your rebuttal. You’ve addressed most of my concerns, so I will raise the score.

---

> > > ### Author Response · Authors · 2025-08-05
> > >
> > > Thank you very much for your positive evaluation! We sincerely appreciate your recognition of both the effectiveness of our proposed method and the overall organization of the paper. Your constructive suggestions have been extremely helpful in improving the clarity of our work. We will revise the manuscript accordingly to eliminate potential confusion and to better illustrate our key contributions through additional case studies. Once again, thank you for your time, effort, and thoughtful feedback throughout the review process!

---

### Official Review · Reviewer_12t9 · 2025-06-28

**Clarity:** 3
**Significance:** 2
**Originality:** 2
**Rating:** 3
**Confidence:** 4

**Summary:**

This paper presents CRAFT (Cross-Attention based Future Link Predictor on Temporal Graphs), an architecture for future link prediction in temporal graphs. The authors identify two critical yet underexplored requirements for this task: incorporating unique node identifiers and enabling target-aware matching between source and destination nodes. Departing from prior approaches that rely on memory and aggregation modules, CRAFT adopts a simpler design composed of two key components: learnable node embeddings and a cross-attention mechanism that aligns candidate destination nodes with the source node’s recent interaction history. This architecture facilitates expressive and adaptive modeling of temporal interaction patterns while maintaining architectural simplicity, enabling effective generalization to both seen and unseen edges.

**Questions:**

None

**Ethical Concerns:**

["NO or VERY MINOR ethics concerns only"]

**Final Justification:**

Thank you for the author's submission. Appreciate the authors’ efforts and contributions. The final rating reflects a comprehensive evaluation of the paper’s strengths and limitations.

**Quality:**

3

**Strengths And Weaknesses:**

This manuscript presents several strengths.

(S1) The writing is clear and easy to follow.

(S2) The proposed architecture is easy to implement and suitable for unseen future links.

(S3) The baselines and datasets are comprehensive.

Nevertheless, this manuscript also has several weaknesses that should be addressed.

(W1) The motivation for unique node identifiers is relatively weak. In real applications of temporal graphs, different nodes should have rich and different features, why these features cannot be used to differentiate nodes?

(W2) I think it is a bit wired that the framework only utilizes learned node embeddings and positional encoding, without utilizing any node features and edge features. Although I noticed that the authors claim many of the benchmark’s datasets lack node features. However, this is not the case in practical scenarios. As the example in your Figure 3, the products ‘iphone/air pods/iphone case’ should have texts, images, and other statistical features if we want to conduct a real next-item- recommendation task. If we assume nodes have no features, how could we know what does a specific node (say node 100) indicate? And how could such a system be useful in practice? Hence, I think the practical utility of the proposed model is rather limited.

(W3) Moreover, how does the model handle newly coming nodes? Since practical temporal graphs will involve not only unseen edges but also unseen nodes as time evolving. The node embeddings are learned during model training. How does the model handle these unseen nodes seems to be a crucial limitation.


(W3) The paper claims that the learnable node embedding will enable powerful expressiveness. However, there lacks a theoretical justification that the embedding could improve expressiveness for the temporal prediction task. Theoretical results of static gnns may not be seamlessly transformed into temporal gnns.

---

> ### Author Rebuttal · Authors · 2025-07-29
>
> Thanks for your valuable time and constructive feedback.
>
> >W1&W2-1. "If we assume nodes have no features, how could we know what does a specific node (say node 100) indicate?"
>
> We use the node identity (e.g., index 100) to represent a node and retrieve its interaction history. For example, $(100, 99, t)$ denotes an interaction between node 100 and node 99 at time $t$. **This is exactly how existing benchmark datasets are structured**, i.e., nodes are treated as ID-based entities, and their interaction patterns are inferred purely from interaction data if no features are available. In fact, **this setup is widely adopted in practical applications.** For example, a large body of work in the recommendation literature, one of the most important applications of future link prediction, focuses on predicting a user's next interaction based solely on interaction data[R5].
>
> ---
> > W1&W2-2. "In real applications of temporal graphs, different nodes should have rich and different features, why these features cannot be used to differentiate nodes?"
>
> We agree that high-quality node features can be invaluable for link prediction tasks if available. However, this is not always the case, and we would like to clarify that link prediction models based solely on interaction data are often required in real-world scenarios and present a valuable research challenge.
>
> 1. **Node features in real-world dynamic graphs are often incomplete, noisy, and difficult to align across node types, particularly in bipartite and heterogeneous graphs.** For example, the GoogleLocal dataset [R1], a real-world dataset of restaurant reviews from Google Maps, includes two node types: users and places, whose features (e.g., user profile vs. place attributes like price) cannot be aligned. Furthermore, many features are missing, like the price feature for over 267,000 out of 267,336 places. These issues explain why many temporal graph benchmarks often lack features. If the features are sparse or of poor quality, they can introduce noise and degrade performance.
>
> 2. **Existing studies have shown that interaction data provides more valuable information than node features for link prediction**, whether in static graphs [R6] or temporal graphs [R1]. CRAFT, which does not use any features, still achieves superior performance compared to the baselines that incorporate features. **This highlights the critical importance of modeling temporal interactions and the limitations of current methods in fully leveraging interaction data.**
>
> 3. **While node features can differentiate nodes, they cannot capture interaction patterns as effectively as learnable node embeddings.** We tested this on the *tgbl-flight* dataset, comparing `CRAFT-core_w_embed` (using node embeddings) with `CRAFT-core_w_feat` (replacing embeddings with features). (`CRAFT-core` refers to CRAFT without the repeated time and last elapsed time encodings, which we removed to mitigate their impact.)
>
>    Table 1. Feature-based CRAFT vs. Embedding-based CRAFT
>    |Dataset|CRAFT-core_w_feat|CRAFT-core_w_embed|
>    |-|-|-|
>    |tgbl-flight|86.36|90.61|
>
>    As noted in TGB [R2], the node features for each airport *tgbl-flight* include attributes like the ISO region code, longitude, and latitude, which can differentiate nodes. However, Table 1 shows that the node embeddings is more effective than node features in CRAFT. This is because node embeddings are learned from the interaction data, which better captures the interaction patterns of nodes.
>
> That said, we recognize that **leveraging node features for link prediction is an important research direction, and we are actively investigating it**. However, addressing this challenge is non-trivial and will require incremental efforts, such as **creating new benchmark datasets with high-quality node attributes** and **developing models that effectively integrate these features**. In this paper, we focus on modeling node interaction patterns using interaction data, as this remains a valuable and unsolved problem.
>
> **To enable CRAFT to leverage node features, we explore incorporating them by modifying node embeddings.** Specifically, we project edge and node features (if available) by a linear layer and concatenate them with the learnable embeddings. The concatenated features are projected to match the dimension of the embeddings. Table 2 shows that `CRAFT_add_feat` performs comparably to the original CRAFT, suggesting that **this method can serve as a basic variant of CRAFT for incorporating features when needed**. While `CRAFT_add_feat` slightly outperforms CRAFT on the wikipedia and reddit datasets, it underperforms on mooc. This reinforces our previous point that node features may often lack high quality or be less valuable for link prediction.
>
> Table 2. CRAFT with Features
> ||wikipedia|reddit|mooc|
> |-|-|-|-|
> |CRAFT_add_feat|**88.40±0.39**|**89.42±0.04**|61.03±0.62|
> |CRAFT_original|88.25±0.26|89.33±0.11|**62.32±0.39**|
>
> ---
> > W3. "how does the model handle newly coming nodes? Since practical temporal graphs will involve not only unseen edges but also unseen nodes as time evolving."
>
> Thank you for this valuable question. As we mentioned in a related context raised by Reviewer Z46w, **Predicting the next interaction for a new node without any historical interactions is considered a *cold-start* problem**, which is distinct from the classic future link prediction task addressed in this paper and related literature—predicting future interactions by modeling node interaction patterns based on historical data. As a result, neither CRAFT nor any of the baselines can handle this cold-start problem.
>
> **The cold-start problem is typically studied separately from classic future link prediction.** For instance, in the recommendation literature, common approaches include learning from metadata of new items [R7] or inferring a new user's preferences by requesting explicit ratings [R3]. Once the new node has accumulated some interactions, recommender systems can switch to classic future link prediction methods to predict subsequent interactions [R7].
>
> The dynamic graph learning community [R1] has also recognized the distinction between the cold-start problem and classic link prediction based on historical interactions. Accordingly, recent benchmark datasets (e.g., the TGB-Seq benchmark used in our paper) are designed to include only test nodes that have appeared in the training set. **Therefore, our focus in this paper is on the classic future link prediction task, specifically addressing the limitation of existing methods in predicting unseen edges.** That said, we acknowledge the importance of the cold-start problem and plan to address it in future work.
>
> ---
> > W4. "...there lacks a theoretical justification that the embedding could improve expressiveness for the temporal prediction task."
>
> Thank you for this insightful question. The expressiveness of temporal GNNs is defined by the ability to map two distinct nodes in a temporal graph to unique embeddings at a given time, i.e., distinguishing between different nodes in a temporal context [R4]. By introducing node embeddings, we provide unique representations for each node, allowing the model to effectively differentiate between nodes based on their embeddings. We will include this discussion in our revision.
>
> [R1] TGB-Seq Benchmark: Challenging Temporal GNNs with Complex Sequential Dynamics. ICLR'25
>
> [R2] Temporal graph benchmark for machine learning on temporal graphs. NeurIPS'23
>
> [R3] Preference elicitation as an optimization problem. RecSys'18
>
> [R4] Provably expressive temporal graph networks. NeurIPS'23
>
> [R5] Neural Collaborative Filtering. WWW'17
>
> [R6] Revisiting Link Prediction: A Data Perspective. ICLR'24
>
> [R7] MARec: Metadata Alignment for Cold-start Recommendation. RecSys'24

---

> ### Author Response · Authors · 2025-08-05
> **Summary and Looking Forward to Further Discussions**
>
> Thank you again for your thoughtful and constructive feedback on our paper. We understand that reviewing many manuscripts requires significant time and effort, and we sincerely appreciate the attention you've given to ours. For your convenience, we've provided a summary of our rebuttal below.
>
> - **How link prediction without features works in practice**: Standard practice treats nodes as identity-based entities, with their semantics inferred from interaction histories. This is consistent with how existing benchmark datasets are structured and is widely adopted in real-world applications such as recommendation systems.
> - **Why link prediction without features is meaningful**:
>   - Real-world features are often incomplete, noisy, or unaligned across node types.
>   - Empirical studies and our experiments show that interaction data is more informative than features for link prediction.
>   - Node features cannot capture interaction patterns as effectively as learnable node embeddings. We validate this through additional experiments on *tgbl-flight*, the only benchmark dataset with node features.
>
> - **How to deal with new nodes**: We clarify that predicting interactions for new nodes without history is a classic *cold-start* problem, which differs from our focus — predicting unseen edges based on historical interactions — a challenge that existing methods still struggle to address.
>
> - **Theoretical justification of node embeddings**: We explain how node embeddings improve the expressiveness of temporal GNNs, supported by prior literature on the expressiveness of temporal graph models.
>
> To summarize, **our key contribution lies in addressing a fundamental limitation of existing temporal GNNs**—their difficulty in predicting unseen edges—which restricts their applicability in real-world tasks that require generalization across both seen and unseen edges. Our method is the first to tackle this challenge directly, using a simple yet effective design that **removes the need for memory and aggregation modules**. We hope this work will **inspire the community to revisit conventional architectures in temporal graph learning**.
>
> While we have not yet explored the cold-start setting and graphs with rich node features, we emphasize that these challenges remain largely open in existing literature. We are actively investigating both directions, and we believe this work provides a solid foundation for future research in these areas.
>
> We greatly appreciate your time and effort in reviewing our submission, and we hope our responses help clarify the design choices and contributions of our work. If you have any further questions or require additional clarification, we welcome continued discussion and engagement.

---

### Official Review · Reviewer_Z46w · 2025-06-30

**Clarity:** 3
**Significance:** 3
**Originality:** 3
**Rating:** 4
**Confidence:** 3

**Summary:**

This paper introduces CRAFT architecture, whose most significant differences from the previous models are (1) the introduction of unique node identifiers and (2) the use of cross attention mechanism (coined as target-aware matching in the paper).
The one of the strongest motivations of this paper is, the memory or aggregation components commonly used in previous work may not be required, if trainable a trainable node embedding unique for each node is used instead.
Given a source node and a set of candidate destinations, it first finds k neighbors for the source node (in the embedding space), sort them by recency, and add trainable positional encodings to node embeddings. Then it applies several rounds of cross attention mechanism, adds elapsed time information, and then finally predicts the score for each candidate destination.
From experiments, it is empirically proven that the suggested CRAFT method outperforms former approaches, while at the same time achieving reduced computational complexity.

**Questions:**

* Is there a way to obtain node embeddings for unseen nodes? I believe the use of node attributes & aggregation in previous work is mainly for this.

**Ethical Concerns:**

["NO or VERY MINOR ethics concerns only"]

**Final Justification:**

I read the authors’ response and other reviews, and decided to keep the score.

**Limitations:**

Clarification & discussion on setups where there exist unseen node and unseen edge would be preferred.

**Quality:**

3

**Strengths And Weaknesses:**

Strengths
* Large performance improvement introduced by simple change
* Paper is organized well and easy to understand
* Analyses are done thoroughly

Weaknesses
* Some details are not clear: e.g. the formulation of TimeProjection function (I assume it's similar to JODIE's though)
* I believe it can't handle when a *node* is not present during training, i.e. inductive setup, which is the scenario often discussed in literature.

---

> ### Author Rebuttal · Authors · 2025-07-29
>
> Thanks for your valuable time and constructive suggestions.
> > w1. "Some details are not clear: e.g. the formulation of TimeProjection function (I assume it's similar to JODIE's though)"
>
> Yes, the **TimeProjection** function is a linear layer that maps the elapsed time $t$ to a vector in $\mathbb{R}^d$ (as noted in Line 151) via $W_t \cdot t$, where $W_t \in \mathbb{R}^{d \times 1}$. We will revise the paper to clarify this formulation.
>
> ---
> > w2. "I believe it can't handle when a node is not present during training, i.e. inductive setup, which is the scenario often discussed in literature."
>
> Thank you for this valuable point. There are two cases where a node is unseen during training but appears in testing:
>
> 1. **Case 1:** The new node has no historical neighbors before the prediction time. **Neither CRAFT nor existing baselines can make accurate predictions for this node**, as all methods rely on modeling interaction patterns based on historical data to predict future neighbors.
>
>     In fact, **predicting the next interaction for a new node with no history is a typical *cold-start* problem**. For example, in the recommendation literature, the cold-start problem is often studied separately [R2], with common approaches like learning from metadata of new items [R3] or inferring a new user's preferences through asking the new user's explicit ratings [R4]. Notably, the dynamic graph learning community [R1] has also recognized the difference between the cold-start problem and classic link prediction based on historical interactions.
>
> 2. **Case 2:** The new node has historical neighbors before the prediction time. **CRAFT can handle Case 2 as existing methods do**, as it only relies on the embeddings of historical neighbors. To empirically evaluate CRAFT on this, we follow the inductive setup from [R5], where test nodes $S'$ are selected, and all edges connected to them are removed from the training set. The model is then trained on the remaining graph and evaluated on the edges involving nodes in $S'$. As shown in Table 1, **CRAFT consistently outperforms these baselines under this inductive setup**.
>
>     Table 1. MRR Results under the Inductive Setup
>     ||CRAFT|JODIE|TGN|DyGFormer|
>     |-|-|-|-|-|
>     |mooc|**62.66**|27.66|46.46|42.29|
>     |lastfm|**46.28**|19.22|26.44|40.20|
>     |GoogleLocal|**55.83**|24.42|45.03|18.52|
>
> In summary, both CRAFT and existing methods focus on link prediction based on historical interaction data, a problem highly relevant to real-world dynamic systems and extensively studied. However, existing methods struggle with predicting unseen edges [R1], a challenge that has not been effectively addressed. In this paper, we prioritize addressing this limitation to enhance the applicability of temporal graph learning methods in real-world applications. While we recognize the importance of the cold-start problem, we plan to explore it in future work.
>
> ---
> > Q1. "Is there a way to obtain node embeddings for unseen nodes? I believe the use of node attributes & aggregation in previous work is mainly for this."
>
> Thank you for this insightful question. One commonly proposed solution [R3] is to incorporate auxiliary node information (e.g., name, description) and learn to project these features into the same embedding space as the node embeddings. However, this approach requires high-quality node features, which are lacking in most existing benchmark datasets. As a result, we cannot verify this method on existing datasets, and previous works also cannot confirm their effectiveness of using node features for link prediction of unseen nodes.
>
> We recognize that **leveraging node features to address cold-start and general link prediction tasks is an important research direction, and we are actively investigating it**. However, addressing this challenge is non-trivial and will require incremental efforts, such as **creating new benchmark datasets with high-quality node attributes** and **developing models that effectively integrate these features**.
>
> ---
> > L1. "Clarification & discussion on setups where there exist unseen node and unseen edge would be preferred."
>
> Thank you for your valuable suggestions. We will add the following discussion in our revision:
>
> Unseen nodes commonly arise in real-world dynamic graphs, such as when a new user joins a social network or a new item is added to an e-commerce platform. In these cases, the new node lacks interaction data, making it difficult to predict its future interactions or preferences. This challenge is often referred to as the cold-start problem, where common approaches include leveraging auxiliary node information (e.g., metadata, descriptions) to predict the behavior of new nodes.
>
> Unseen edges are prevalent in dynamic graphs, as new interactions continually emerge. For instance, a user may purchase an item for the first time or follow another user they have not interacted with before. Predicting such unseen edges is challenging, and existing methods often struggle in this area. In our work, we address both unseen and seen edge prediction, filling a critical gap in the existing literature, which typically focuses only on seen edges.
>
> We hope this clarification enhances the understanding of our approach. If you have any further suggestions, please let us know.
>
>
> [R1] TGB-Seq Benchmark: Challenging Temporal GNNs with Complex Sequential Dynamics. ICLR'25
>
> [R2] Cold-start Recommendation by Personalized Embedding Region Elicitation. UAI'24
>
> [R3] MARec: Metadata Alignment for Cold-start Recommendation. RecSys'24
>
> [R4] Preference elicitation as an optimization problem. RecSys'18
>
> [R5] Towards Better Dynamic Graph Learning: New Architecture and Unified Library. NeurIPS'23

---

> ### Author Response · Authors · 2025-08-05
> **Summary and Looking Forward to Further Discussions**
>
> Thank you again for your thoughtful and constructive feedback on our paper. We understand that reviewing many manuscripts requires significant time and effort, and we sincerely appreciate the attention you've given to ours. For your convenience, we've provided a summary of our rebuttal below.
>
> - **TimeProjection function**: We will revise the paper to clarify the formulation of the TimeProjection function.
>
> - **How to handle inductive setup**: We distinguish between two cases:
>   - **New nodes with no historical neighbors (cold-start problem)**: Neither CRAFT nor existing baselines can handle this, as the task requires auxiliary information. We discuss related solutions in the literature and plan to explore this in future work.
>   - **New nodes with historical neighbors**: CRAFT can handle this case. We followed the inductive setup from existing literature and evaluated CRAFT on mooc, lastfm, and GoogleLocal, showing that it consistently outperforms strong baselines under this setting.
> - **How to obtain node embeddings for unseen nodes**: Generating node embeddings for unseen nodes without history is a cold-start problem. Prior work in cold-start literature suggests projecting node features into the same latent space as learnable embeddings trained on seen nodes. We acknowledge the importance of leveraging node features to address the cold-start problem and are actively exploring this direction. However, most existing benchmark datasets lack node features, so incremental efforts—such as first creating datasets with high-quality features—are necessary.
> - **Clarification on unseen nodes and unseen edges**: We will add discussion in the revision to explicitly describe the scenarios where unseen nodes and unseen edges occur.
>
> To summarize, **our key contribution lies in addressing a fundamental limitation of existing temporal GNNs**—their difficulty in predicting unseen edges—which restricts their applicability in real-world tasks that require generalization across both seen and unseen edges. Our method is the first to tackle this challenge directly, using a simple yet effective design that **removes the need for memory and aggregation modules**. We hope this work will **inspire the community to revisit conventional architectures in temporal graph learning**.
>
> We greatly appreciate your time and effort in reviewing our submission, and we hope our responses help clarify the design choices and contributions of our work. If you have any further questions or require additional clarification, we welcome continued discussion and engagement.

---

### Official Review · Reviewer_HCZc · 2025-06-30

**Clarity:** 3
**Significance:** 2
**Originality:** 2
**Rating:** 4
**Confidence:** 3

**Summary:**

The paper introduces a novel method to work on link prediction in dynamic graphs, featuring two components: learnable node embeddings and a "target-aware matching", relying on a cross-attention mechanism.

**Questions:**

- the sampling of negative items is a subject in itself in the Recommender System community. It is shortly/vaguely described here (l262-263). What does "We adopt the negative samples from the benchmarks" mean? Is the random selection focusing on a node, or is it global?
- there are imprecisions in the formalism: the graph model looks like a "link stream" model. Is it exactly one, or not? Also, "since both pairs share two common neighbors" (l178): in a graph, nodes usually have neighbors, not edges.
- "repeat time" is an important concept in your work, however not defined. What does it mean? What happens to "repeat time" if an edge appears irregularly?
- at the core of the model, there is the hyperparameter k, the size of a candidate set. The value of k is not evaluated, however it should have a tremendous impact on the CRAFT's performance. It seems with the sentence l181 that [4] and [36] deal with experiments on this, but it should probably be expanded. Could you explain what effect you expect k to have?

**Ethical Concerns:**

["NO or VERY MINOR ethics concerns only"]

**Final Justification:**

updated score after discussion with authors and clarifications.

**Limitations:**

yes

**Quality:**

2

**Strengths And Weaknesses:**

The paper clearly exposes its aim, and the solution they bring to existing litterature. However, I am afraid that there is a deep problem plaguing the whole paper: authors deal with a problem which is not (exactly) the same as the one presented in the existing litterature. They perform and improve link prediction _for a given node_, while other methods analyse a whole graph and propose to predict links _at the graph level_, which is very different.
Second problem (related to the first one): while authors mention in 4.2.2 (and Fig3) the use case of a recommender system scenario, they completely neglect this litterature, while it would clearly be more relevant to their problem, since recommendation focuses on a _candidate user_ with a potential set of items. There are already a large body of works on recommending items in a dynamic setting, with or without graph learning (and with several attempts to encode time as presented here, see Sequential Recommendation¹). But TGN, TGAT and others (JODIE, DyRep) were not meant to work by focusing on a user/node as it is done here.

¹: note that in a recommender system, the candidate set usually is the entire collection of items

In light of this, all the paper is affected: the Related Work section is far too short or out of scope (1st paragraph), Experiments confront CRAFT to irrelevant methods (or omit relevant baselines), etc.

Also, in 4.2.1, it is claimed that GNNs are plagued by "lack of distinctive node embeddings", sustained by a 2020 paper. The 5-year gap is important here, as most GNNs have been using random/other initial embeddings for years.

Minor remarks.

Bibliography:
- "Attention is all you need" as more than one author…
- use {} to prevent capital letters from being transformed to lowercase (eg, gnns)
- BPR was published, don't cite arxiv version when possible

---

> ### Author Rebuttal · Authors · 2025-07-29
>
> Thanks for your valuable time and constructive suggestions.
>
> > W1. "authors deal with a problem which is not (exactly) the same as the one presented in the existing literature. They perform and improve link prediction for a given node, while other methods analyse a whole graph and propose to predict links at the graph level, which is very different."
>
> The problem we address is indeed the same as that presented in the existing literature, such as in TGN and JODIE. As stated in TGN: "In future edge prediction, the goal is to predict the probability of an edge occurring between two nodes at a given time", which is exactly the task that CRAFT performs — predicting the probability of an edge given its two terminals and a timestamp.
>
> We are not entirely sure what is meant by "other methods...predict links at the graph level," but we would like to offer the following clarifications that may help address any misunderstandings:
>
> 1. **The inputs and outputs are the same for all baselines and CRAFT**. The inputs consist of the time $t$ and nodes $s,d$, and the output is the probability of the edge $(s,d)$ occurring at time $t$. **All baselines, including CRAFT, cannot output the next edge $(s,d)$ given only the time $t$**.
> 2. **The focus of all baselines and CRAFT is the same**: modeling the interaction patterns of a **node** using its historical neighbors. For example, recent works [R3] apply a Transformer encoder *on the sequence of historical neighbors* to obtain node representations and then predict the edge score. CRAFT is similar to these works, inferring the node's interaction patterns based on its historical neighbors.
> 3. Please distinguish between **continuous-time dynamic graph (CTDG)** and **discrete-time dynamic graph (DTDG)**. Methods usually process each static graph as a whole separately in a DTDG and then model these graphs as a sequence. However, CRAFT and baselines are all for CTDGs, where the graph is modeled as a stream of edges.
> ---
> > W2. "while authors mention...recommender system scenario, they completely neglect this literature, while it would clearly be more relevant to their problem..."
>
> We would like to clarify the relationship between our focus (future link prediction in temporal graph) and the recommendation literature:
> 1. **Recommendation is often considered a special application scenario for future link prediction**, as stated in works like JODIE and TGN. This is also why many existing datasets used in temporal graph learning come from recommendation systems, such as lastfm (from JODIE for music recommendation) and tgbl-review (from TGB [R1], a dataset for e-commerce recommendation curated from Amazon.com).
> 2. **Temporal graph learning covers a wider range of application scenarios**, from education (the mooc dataset from JODIE), to transportation (tgbl-flight [R1]), and transactions (tgbl-coin [R1]). In essence, temporal graph learning aims to capture a variety of evolving patterns in real-world dynamic systems, while recommendation is one specific use case.
> 3. **Recommendation focuses on bipartite graphs** (user-item interactions), while **temporal graph learning aims to work on general temporal graphs**, including both bipartite and non-bipartite graphs.
> 4. **Experimental comparison with recommendation methods in Appendix B.8 demonstrates that CRAFT outperforms widely recognized recommendation methods**: Most of temporal graph learning literature (e.g., JODIE and TGN) does **not** specifically review or compare with recommendation literature due to their differences. That said, we recognize the value of recommendation methods as strong baselines, especially in bipartite graph settings. Therefore, we've **compared CRAFT with two recommendation baselines, SASRec and SGNN-HN** in Appendix B.8 and the results demonstrate CRAFT's superiority.
> 5. **We will add a review of recommendation literature to our revised manuscript** and provide further clarification of the differences between recommendation and dynamic graph learning. If you have any specific works that you believe should be included or compared in our review, please feel free to suggest them.
>
> ---
> > W3. "it is claimed that GNNs are plagued by "lack of distinctive node embeddings", sustained by a 2020 paper. The 5-year gap is important here, as most GNNs have been using random/other initial embeddings for years."
>
> Thanks for this valuable point. Our emphasis is that dynamic graph learning methods have not fully embraced the idea of node identifiers as static GNNs have. Many still rely on zero features when explicit features are absent, limiting their expressiveness—this is exactly what motivates the use of learnable node embeddings in our CRAFT.
>
> ---
> > W4. Minor remarks
>
> Thank you for your careful reading and helpful feedback! We will revise the bibliography accordingly.
>
> > Q1. "What does "We adopt the negative samples from the benchmarks" mean? Is the random selection focusing on a node, or is it global?"
>
> For datasets from the TGB and TGB-Seq benchmarks, negative samples are provided by the benchmarks themselves: The TGB benchmark samples 100 or 20 negative destinations for each source node from historical and random negative edges [R1], while the TGB-Seq benchmark randomly samples 100 negatives for each source [R2]. For other datasets without publicly available negatives, we follow the approach outlined in [R2] to randomly select 100 negative destinations for each source node.
>
> > Q2-1. "the graph model looks like a 'link stream' model"
>
> Yes, we focus on continuous-time dynamic graphs, where the graph is modeled as a stream of edges, as defined in most existing literature (JODIE, TGN, DyGFormer[R3], etc.).
>
> > Q2-2. "Also, 'since both pairs share two common neighbors' (l178): in a graph, nodes usually have neighbors, not edges."
>
> To clarify, in this sentence, we are referring to the nodes b and a, and b and f sharing common neighbors, respectively, rather than saying that the two edges (b, a) and (b, f) share common neighbors. We will revise the sentence for better clarity: "since both **b and a**, and **b and f** share two common neighbors, c and d, and d and e, respectively".
>
> > Q3. "'repeat time'.. What does it mean?"
>
> Repeat time refers to the number of times an edge appears in the dynamic graph upon the prediction time $t$. Previous studies [R2, R4] have shown that some datasets contain excessive repeated edges, and most methods tend to excel only at predicting these repeated edges. **CRAFT is the first model to effectively address both repeated edges (seen) and new edges (unseen)**, which is crucial for real-world dynamic graph applications where unseen edges are common.
>
> ---
> > Q4. "k, the size of a candidate set...Could you explain what effect you expect k to have?"
>
> To clarify, $k$ is **not** the size of the candidate set but rather refers to the number of hops of neighbors ($k$-hop neighbors) considered in existing methods, as mentioned in l175. CRAFT does not consider multi-hop neighbors, i.e., $k = 1$, and outperforms baselines that consider multi-hop neighbors.
>
> The size of the candidate set (denoted as $q$) does not affect the prediction results of CRAFT or other baselines, but it does impact the evaluation score. **A larger candidate set increases the evaluation difficulty and typically lowers the MRR scores**, as the methods must rank the positive destination among a larger pool of candidates.
>
> To assess the robustness of CRAFT, we conducted experiments with $q=500$ on the lastfm and wikipedia datasets. As shown in Table 1, CRAFT either outperforms or performs comparably to the best baseline when $q=500$. Notably, the significant superiority of CRAFT over the baselines on lastfm with $q=100$ is also maintained with $q=500$, further demonstrating CRAFT's robustness.
>
> Table 1: MRR with candidate set size $q=500$
> |Dataset|CRAFT|DyGFormer|GraphMixer|
> |-|-|-|-|
> |lastfm|**28.56**|21.20|10.91|
> |wikipedia|69.08|**70.11**|49.98|
>
> ---
> We greatly appreciate your valuable time and insightful comments! Should you have any further questions, we would be happy to discuss further.
>
> [R1] Temporal graph benchmark for machine learning on temporal graphs. NeurIPS'23
>
> [R2] TGB-Seq Benchmark: Challenging Temporal GNNs with Complex Sequential Dynamics. ICLR'25
>
> [R3] Towards Better Dynamic Graph Learning: New Architecture and Unified Library. NeurIPS'23
>
> [R4] Towards Better Evaluation for Dynamic Link Prediction. NeurIPS'22
>
> [R5] Do we really need complicated model architectures for temporal networks? ICLR'23

---

> > ### Comment · Reviewer_HCZc · 2025-08-06
> >
> > I thank the authors for their responses.
> >
> > I hope that all of the clarifications from the authors will be included in the revised manuscript (Q3 "repeat time", Q4 "candidate set notation", Q2-1 the expression "link stream", Q1 a short discussion on the negative sampling). I have no further suggestion on the recommendation part, but I like that a mention is added in the revised text.
> >
> > I will raise my current score, but I still find that the paper should have a clearer writing.

---

> > > ### Author Response · Authors · 2025-08-07
> > >
> > > Thank you so much for your positive feedback! We greatly appreciate your thoughtful review and are fully committed to making the necessary revisions to further improve our paper. Our detailed revision plan is as follows:
> > >
> > > 1. **Q3 "repeat time"**: We will introduce the notation for repeat time, $\mathcal{R}(s,d,t)$, in Line 156 and clarify that $\mathcal{R}(s,d,t)$ represents the number of times node $s$ has interacted with node $d$ before time $t$.
> > > 2. **Q4 "candidate set notation"**: Since the size of the candidate set does not affect model predictions and is only discussed in the time complexity analysis in Section 4.3, we've decided to introduce the concept of the candidate set exclusively in Section 4.3 to avoid any confusion. Additionally, we will include a notation table in the appendix to define all terms used throughout the manuscript.
> > > 3. **Q2-1 "the graph model looks like 'link stream'"**: We currently define the graph model as $G = (V, E)$, where $V$ is the set of nodes and $E$ is a sequence of edges ordered by non-decreasing timestamps, i.e., $E={e_1, e_2, \dots, e_m}$, with each edge $e_i = (s_i, d_i, t_i)$ as described in Line 100. To eliminate any potential confusion, we will clarify the distinction between continuous-time and discrete-time dynamic graphs in the revised manuscript.
> > > 4. **Q1 "negative sampling"**: We will include an explanation of negative sampling in Line 263, under the "Experimental Settings" section, as mentioned in our response to Q1.
> > > 5. **"recommendation literature"**: We will emphasize that recommendation is a specific application within the broader field of future link prediction in Section 2 ("Related Work"). Furthermore, we will provide a more detailed review of the recommendation literature, particularly the recommendation methods used in our supplementary experiments, in the appendix.
> > >
> > > We were pleased to see that the reviewer acknowledged that "*The paper clearly exposes its aim, and the solution they bring to existing literature*" in the previous review. We interpret this as an indication that **the core structure and clarity of our contribution are well-received. With the revisions outlined above, we believe the manuscript will be clearer and more accessible.**
> > >
> > > Once again, thank you for your constructive feedback. **We sincerely hope that our work will facilitate the practical application of temporal graph learning by addressing both seen and unseen edges, and inspire the community to revisit the conventional memory and aggregation architectures in this field.** Should you have any additional suggestions or concerns, we would be happy to engage in further discussion.

---

> > > > ### Comment · Reviewer_HCZc · 2025-08-07
> > > >
> > > > 3. It seems that you are reluctant to use the expression "link stream". I don't understand why, it would help readers make the link (no pun intended) with other papers using the same formalism.
> > > >
> > > > OK for the rest, thank you for the active & fruitful discussion.

---

> ### Author Response · Authors · 2025-08-05
> **Summary and Looking Forward to Further Discussions**
>
> Thank you again for your thoughtful and constructive feedback on our paper. We understand that reviewing many manuscripts requires significant time and effort, and we sincerely appreciate the attention you've given to ours. For your convenience, we've provided a summary of our rebuttal below.
>
> - **Is the problem the same as in prior literature?**: The task we address is consistent with existing literature on future link prediction, specifically in works such as TGN and JODIE. We also clarify that CRAFT, similar to the baselines, focuses on predicting edge probabilities based on historical interactions, rather than making graph-level predictions.
>
> - **Recommendation literature comparison**:
>   - Temporal graph learning spans a broader range of applications, with recommendation being one of the many use cases.
>   - Temporal graph learning focuses on general temporal graphs, whereas recommendation typically deals with bipartite graphs.
>   - We have already compared CRAFT with recommendation baselines, demonstrating its superior performance.
>   - In the revised manuscript, we will include a detailed review of the recommendation literature.
>
> - **The effect of candidate set size**: A larger candidate set increases the evaluation difficulty and typically lowers the MRR scores. To demonstrate the robustness of CRAFT, we conducted additional experiments with a candidate set size of 500. Larger candidate sets are not feasible due to the substantial increase in inference time; for instance, CAWN took over 80 hours to complete the experiment with 500 candidates.
>
> - **Clarification on specific terms**: We have clarified the method of generating negative samples, the definition of repeat time, the graph model, and other potential sources of confusion.
>
> To summarize, **our key contribution lies in addressing a fundamental limitation of existing temporal GNNs**—their difficulty in predicting unseen edges—which restricts their applicability in real-world tasks that require generalization across both seen and unseen edges. Our method is the first to tackle this challenge directly, using a simple yet effective design that **removes the need for memory and aggregation modules**. We hope this work will **inspire the community to revisit conventional architectures in temporal graph learning**.
>
> We greatly appreciate your time and effort in reviewing our submission, and we hope our responses help clarify the design choices and contributions of our work. If you have any further questions or require additional clarification, we welcome continued discussion and engagement.

---

> ### Author Response · Authors · 2025-08-07
>
> Haha, we sincerely appreciate your keen observation! After reviewing several prior papers, including JODIE, TGN, TGAT, GraphMixer, and DyGFormer, we noticed that the term "link stream" was not used in any of them. Most of these studies describe continuous-time dynamic graphs as "a sequence of temporal interactions." For instance, JODIE refers to "an ordered sequence of temporal user-item interactions," and TGN describes their model as "a sequence of time-stamped events." This led us to adopt the phrasing of "a sequence of edges ordered by non-decreasing timestamps" to maintain consistency with existing literature.
>
> That said, we did come across the term "link stream" in the paper "Combining structural and dynamic information to predict activity in link streams" (ASONAM’17) [R1], which uses it within the context of temporal graphs. To make the manuscript more accessible for readers familiar with the term, we'll revise the description as follows:
>
> We define a temporal graph as $G = (V, E)$, where $V$ is the set of nodes and $E$ is a sequence of edges ordered by non-decreasing timestamps, i.e., $E={e_1, e_2, \dots, e_m}$, with each edge $e_i = (s_i, d_i, t_i)$. This is also known as a link stream graph model [R1].
>
> If you have any further suggestions or references regarding the term "link stream", please feel free to share them with us. Thank you again for your professional and thoughtful feedback. It’s a pleasure to have you as a reviewer and to engage in this discussion with you.
>
> [R1] Combining structural and dynamic information to predict activity in link streams. ASONAM'17

---

> > ### Comment · Reviewer_HCZc · 2025-08-08
> >
> > You are right, the "link stream" expression is not that frequent, perhaps I should not have insisted on that. However, the formalism is rather frequent. Especially, I remembered a talk about "link streams", but looking back at the slides, the precise expression used was "streaming graph", there are several terms for the same idea. "Streaming graph" yields some papers on Google scholar, but most of them are pre-2020 (eg, Streaming Graph Neural Networks Yao Ma, Ziyi Guo, Zhaocun Ren, Jiliang Tang, Dawei Yin SIGIR 2020).
> >
> > As for the "link stream" reference, I think it would be best to cite "Stream graphs and link streams for the modeling of interactions over time" M Latapy, T Viard, C Magnien Social Network Analysis and Mining, 2018 (published after the ASONAM17 but it is a 50-page reference on the model).

---

> > > ### Author Response · Authors · 2025-08-08
> > >
> > > Thank you so much for your detailed follow-up and helpful pointers to additional references! The paper "Stream graphs and link streams for the modeling of interactions over time" indeed provides a comprehensive definition of link streams, and we will cite it in our revision. Additionally, we've found that "Streaming Graph Neural Networks" aligns closely with the graph model in our paper and related works, so we will include it as well to make our paper more complete and accessible.
> > >
> > > We sincerely appreciate your thoughtful feedback and the constructive exchange—it's been a pleasure to discuss these details with you. Should you have any further suggestions or ideas, we would be delighted to continue the discussion.

---

### Decision · Program_Chairs · 2025-09-17

**Decision:**

Accept (poster)

**Comment:**

The paper introduces a novel approach, CRAFT (Cross-Attention-based Future Link Predictor), to temporal graph link prediction, which forgoes the use of memory and aggregation modules commonly used in prior models. Instead, it leverages two key components: learnable node embeddings and a cross-attention mechanism for target-aware matching between source and destination nodes. The proposed method is simple and effective, consistently outperforming a wide range of existing models across 17 different datasets. A primary concern raised by several reviewers is the paper’s handling of cold-start scenarios (where new nodes appear without interaction history). While the authors clarify that CRAFT and existing models cannot handle cold-start problems effectively, a more in-depth exploration or potential solution for these scenarios would have made the paper more comprehensive. In practical applications, such cases are frequent, and addressing them could further solidify the utility of the method.